# Pathogenesis of hypertension in a mouse model for human *CLCN2* related hyperaldosteronism

Corinna Göppner[1,2,8], Ian J. Orozco[1,2,8], Maja B. Hoegg-Beiler[1,2], Audrey H. Soria [1,2], Christian A. Hübner[3], Fabio L. Fernandes-Rosa[4,5], Sheerazed Boulkroun [4,5], Maria-Christina Zennaro[4,5,6] & Thomas J. Jentsch [1,2,7]

Human primary aldosteronism (PA) can be caused by mutations in several ion channel genes but mouse models replicating this condition are lacking. We now show that almost all known PA-associated *CLCN2* mutations markedly increase ClC-2 chloride currents and generate *knock-in* mice expressing a constitutively open ClC-2 Cl$^-$ channel as mouse model for PA. The *Clcn2*[op] allele strongly increases the chloride conductance of zona glomerulosa cells, provoking a strong depolarization and increasing cytoplasmic Ca$^{2+}$ concentration. *Clcn2*[op] mice display typical features of human PA, including high serum aldosterone in the presence of low renin activity, marked hypertension and hypokalemia. These symptoms are more pronounced in homozygous *Clcn2*[op/op] than in heterozygous *Clcn2*[+/op] mice. This difference is attributed to the unexpected finding that only ~50 % of *Clcn2*[+/op] zona glomerulosa cells are depolarized. By reproducing essential features of human PA, *Clcn2*[op] mice are a valuable model to study the pathological mechanisms underlying this disease.

[1] Leibniz-Forschungsinstitut für Molekulare Pharmakologie (FMP), Berlin, Germany. [2] Max-Delbrück-Centrum für Molekulare Medizin (MDC), Berlin, Germany. [3] Institut für Humangenetik, Universitätsklinikum Jena, Jena, Germany. [4] INSERM, UMRS_970, Paris Cardiovascular Research Center, Paris, France. [5] Université Paris Descartes, Sorbonne Paris Cité, Paris, France. [6] Assistance Publique-Hôpitaux de Paris, Hôpital Européen Georges Pompidou, Service de Génétique, Paris, France. [7] NeuroCure Cluster of Excellence, Charité Universitätsmedizin Berlin, Berlin, Germany. [8] These authors contributed equally: Corinna Göppner, Ian J. Orozco. Correspondence and requests for materials should be addressed to T.J.J. (email: Jentsch@fmp-berlin.de)

With an estimated prevalence of about 5–10% in hypertensive patients, the most common cause of secondary arterial hypertension is primary aldosteronism (PA)[1,2]. Aldosterone is synthesized from the precursor cholesterol in zona glomerulosa (ZG) cells in the adrenal cortex in an enzymatic cascade that involves aldosterone synthase (CYP11B2) as the final step[1,3]. Aldosterone production is normally tightly regulated by angiotensin II, extracellular $K^+$ concentration, and ACTH, but is also influenced by other factors[1,3]. Acute stimulation of aldosterone synthesis involves phosphorylation of StAR. It enhances the transport of cholesterol into mitochondria where it is converted to pregnenolone, a precursor for both aldosterone and cortisol in humans or corticosterone in rodents (both synthesized in the adrenal zona fasciculata (ZF)). Medium- to long-term stimulation of aldosterone production is achieved by transcriptional upregulation of aldosterone synthase which is specifically expressed in ZG cells. Being hydrophobic, aldosterone leaves the cell by diffusing across the plasma membrane. Aldosterone exerts its effect on blood pressure (BP) primarily by stimulating $Na^+$ reabsorption in the distal renal tubule through the $Na^+$ channel ENaC. Via electrostatic coupling, increased $Na^+$ absorption augments distal tubular $K^+$- and $H^+$-secretion[4]. Besides kidney, aldosterone acts on several other target organs[1,5].

PA is caused by autonomous aldosterone production in the adrenal cortex[1]. It is associated with arterial hypertension, low plasma levels of renin, the regulatory enzyme that cleaves the angiotensin precursor angiotensinogen, and often with low serum concentration of $K^+$. The high serum aldosterone in patients with PA causes tissue damage in the cardiovascular system and kidney in excess of that caused by high BP alone[1,6–9].

PA has a strong genetic component. Both somatic mutations in aldosterone-producing adenomas and germ-line mutations in Mendelian forms of the disease have been found. The majority of genes mutated in PA encode ion channels and transport ATPases[10,11]. A rise in the cytoplasmic $Ca^{2+}$ concentration ($[Ca^{2+}]_i$) is the most common mechanism by which such mutations stimulate aldosterone production. An increase in $[Ca^{2+}]_i$ can be a direct consequence of gain-of-function mutations in the $Ca^{2+}$ channel genes CACNA1D[12,13] and CACNA1H[14,15], encoding α-subunits of L-type and T-type $Ca^{2+}$ channels, respectively, or of loss-of-function mutations in ATP2B3 (ref. [16]) that encodes the $Ca^{2+}$-extruding plasma membrane ATPase PMCA3. Mutations in other ion-transporting proteins increase $[Ca^{2+}]_i$ indirectly by depolarizing ZG cells to voltages that open voltage-gated $Ca^{2+}$-channels. These include mutations in ATP1A1 (refs. [13,16]), coding for the α1-subunit of the $Na^+,K^+$-ATPase, and mutations in KCNJ5 (ref. [17]) encoding the Kir3.4 (GIRK4) $K^+$ channel. Gain-of-function KCNJ5 mutations typically cause depolarizing $Na^+$ influx by changing the channel's selectivity filter[17]. In mice, disruption of both Kcnk3 and Kcnk9, encoding the "two pore" $K^+$ channels TASK1 and TASK3, respectively, elicits symptoms resembling PA[18–21]. These two "background" $K^+$ channels mediate a large part of the resting conductance that keeps ZG cell plasma membranes hyperpolarized. Most recently, we[22] and others[23] reported that gain-of-function mutations in the CLCN2 gene, which encodes the $Cl^-$ channel ClC-2, underlie early-onset or familial forms of PA.

ClC-2 is a plasma membrane chloride channel that belongs to the CLC family of $Cl^-$ channels and transporters[24,25]. ClC-2 is widely expressed across mammalian tissues[26]. It is only partially open under resting conditions and slowly activates upon hyperpolarization[26]. It is also activated by cell swelling[27] and mildly acidic extracellular pH[28]. Our laboratory previously mapped structures important for these modes of activation to a ~45 residue-long stretch of the ClC-2 amino-terminus[27] and to an intracellular loop between intramembrane helices J and K[28]. Deletions or mutations in these "inactivation" or "gating-modulating" regions abolish the voltage-, pH-, and swelling-sensitivity of ClC-2 and strongly increase current amplitudes. In glia, ClC-2 associates with the cell adhesion molecule GlialCAM which directs ClC-2 to certain cell−cell contacts and "opens" ClC-2 similar to mutations in the "inactivation" domains[29,30].

Several physiological roles of ClC-2 became apparent from the analysis of ClC-2 knock-out (KO) mice ($Clcn2^{-/-}$ mice) and human inherited disease. $Clcn2^{-/-}$ mice display early postnatal testicular and retinal degeneration[31] and slowly progressive leukodystrophy[30,32]. These pathologies were attributed to impaired extracellular ion homeostasis[30–32]. Patients with CLCN2 loss-of-function mutations likewise display leukodystrophy[33] that is sometimes associated with impaired vision and reduced male fertility[34]. Reduced colonic $Cl^-$ reabsorption in $Clcn2^{-/-}$ mice additionally indicates a role in transepithelial transport[35,36].

Here we find that almost all known PA-associated CLCN2 mutations markedly increase ClC-2 $Cl^-$ currents. The only exception likely represents a silent polymorphism. To test the hypothesis that an increase in ClC-2 currents suffices to cause PA, we generate knock-in mice carrying a Clcn2 mutation ($Clcn2^{op}$) that is not found in human PA, but increases ClC-2 currents to a similar degree as the majority of PA-causing human mutations. ZG cells of $Clcn2^{op}$ mice are depolarized by this gain-of-function mutant to an extent that was amply sufficient to open voltage-gated $Ca^{2+}$ channels. The resulting increase in intracellular $Ca^{2+}$ concentration enhances transcription of aldosterone synthase, increases serum aldosterone levels, and causes hypertension, hypokalemia, and mild renal abnormalities. Adrenal cortex-specific disruption of the $Clcn2^{op}$ allele indicates that its effect on aldosterone is ZG cell-intrinsic. Thus, $Clcn2^{op}$ mice reproduce the clinical and biochemical features of human PA.

## Results

**PA-associated CLCN2 mutations increase $Cl^-$ currents.** Until now, six different mutations affecting the ClC-2 $Cl^-$ channel[26] have been identified in patients with PA[22,23] (Fig. 1a). Three mutations, p.Met22Lys (M22K), p.Gly24Asp (G24D), p.Tyr26Asn (Y26N) affect residues in a short segment of the ClC-2 amino-terminus. p.Arg172Gln (R172Q) changes an amino acid at the cytoplasmic end of helix D. p.Lys362Δ (ΔK362) deletes a lysine at the end of the cytoplasmic loop between helices J and K and Ser865Arg (S865R) alters a residue close to the end of the carboxy-terminus. Intriguingly, the N-terminal mutations fall into a highly conserved region we had identified previously as being crucial for channel gating[27], and K362 localizes to the end of a cytoplasmic loop that similarly affects gating[28]. Deletions and missense mutations in either region strongly "open" the ClC-2 channel when expressed in Xenopus oocytes. Instead of being slowly activated by inside-negative voltages, cell swelling, or acidic extracellular pH ($pH_o$), mutant channels are insensitive to these stimuli and current amplitudes are increased more than 10-fold at physiological voltages[27,28]. This suggests that these four mutations identified in PA may strongly increase ClC-2 currents.

Indeed, we previously reported a striking increase in ClC-2 currents with the G24D mutant, both in two-electrode voltage clamp (TEVC) of Xenopus oocytes and in perforated-patch measurements of transfected H295R-S2 adrenocortical cells[22]. No comparable data are available for the mutations described by Scholl et al.[23] who analyzed these mutations using whole-cell patch clamping of transfected cells. However, this procedure, possibly by dialyzing the cell interior, almost completely abolishes the large increase of ClC-2 currents caused by N-terminal mutations[25,37,38]. Accordingly Scholl et al.[23] detected only modest

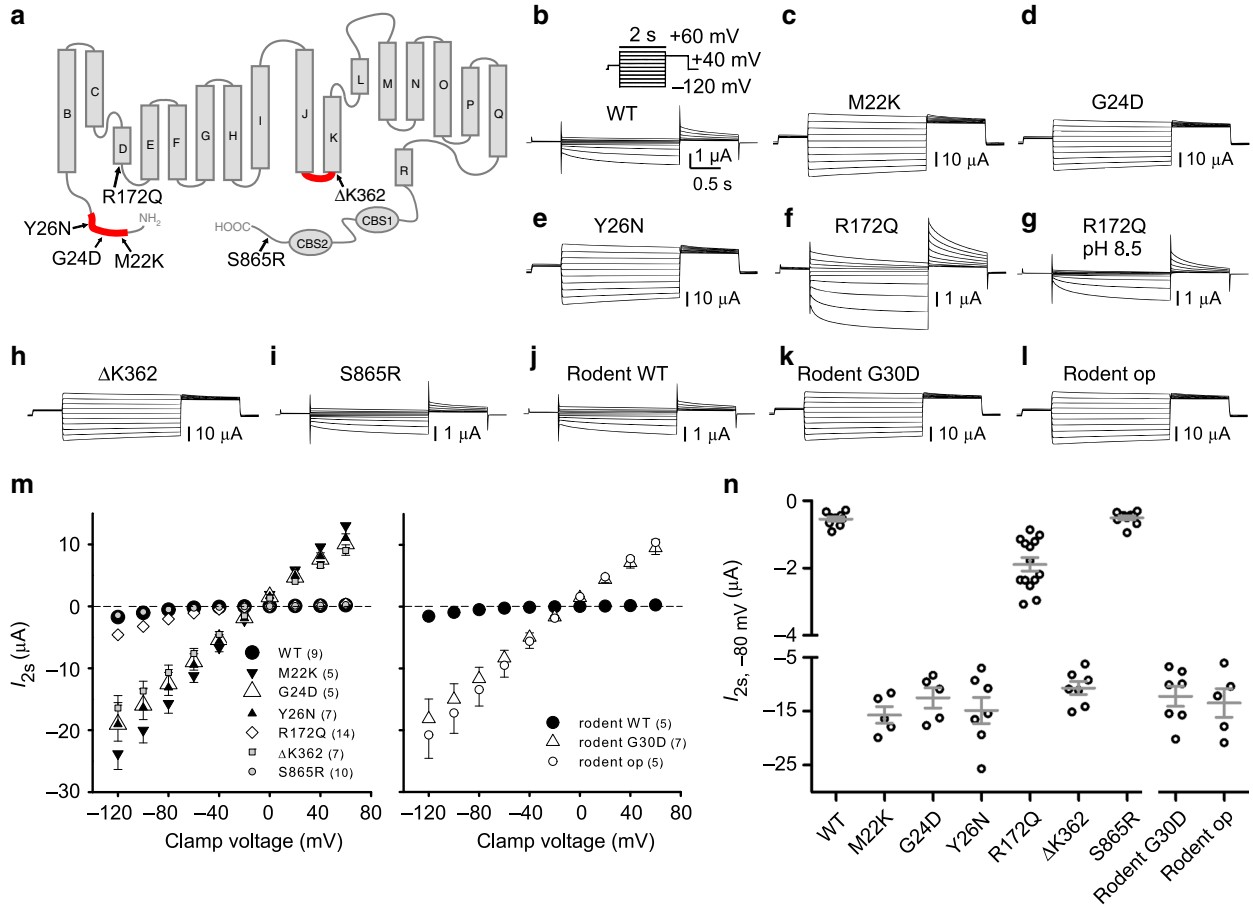

**Fig. 1** Characterization of human *Clcn2* gene variants in primary aldosteronism and the op allele. **a** ClC-2 topology model, based on the CLC crystal structure[66], mapping positions of genetic variants found in early-onset primary aldosteronism[22,23]. N-terminal and J-K linker domains previously shown[27,28] to be important for opening ClC-2 are highlighted in red. **b**–**l** Representative current traces obtained by two-electrode voltage clamp of *Xenopus* oocytes injected with the indicated ClC-2 cRNAs. Voltage clamp protocol indicated in the inset of panel **b**. The rodent "op" mutant exactly mimics the channel encoded by the *Clcn2*op allele in the present *knock-in* mouse model. For comparison to the human G24D mutant, the equivalent G30D rodent mutant was measured. In some experiments (**f**, **g**) individual ClC-2(R172Q)-expressing oocytes ($n = 7$) were first measured with ND109, pH 7.4 and then in ND109 buffered to pH 8.5. **m** Plot of mean current measured at 2 s after clamping to indicated voltages and (**n**) individual currents at a clamp of −80 mV for experiments of **b**–**l**. The number of cells measured is indicated in parenthesis. **m**, **n** Currents are presented as mean ± SEM

changes in gating and have probably overlooked the physiologically relevant effects.

We now evaluated all described PA-related *CLCN2* mutations[22,23] under appropriate conditions in both *Xenopus* oocytes (Fig. 1b–n) and HEK cells (Supplementary Fig. 1b–h). We compared the results to those obtained with a previously described "open" deletion mutant that removes residues Y[32]TQELGAF from the N-terminus of rodent ClC-2 (ref. [27]). Since rodent Y32 corresponds to human Y26, this deletion eliminates the residue changed in the Y26N mutant[23].

When measured by TEVC in *Xenopus* oocytes, all four human mutations in either gating-modulating region[27,28] (M22K, G24D/G30D in rodent, Y26N, and ΔK362) increased ClC-2 currents by an order of magnitude and abolished the strong voltage-dependence of wild-type (WT) ClC-2 (Fig. 1b–e, h, k–n). Currents from these mutants and the N-terminal rodent deletion construct (op) were indistinguishable. By contrast, currents from the R172Q ClC-2 mutant were more moderately increased (~4-fold; Fig. 1f, m, n). Unlike G24D[22] and other N-terminal and J-K loop mutants[28], R172Q partially retained voltage-dependent gating and responded to changes in pH_o (Fig. 1g). The carboxy-terminal S865R variant[23] lacked discernable effects on ClC-2 currents (Fig. 1i, m, n), suggesting that it represents a silent polymorphism.

Similar results were obtained with HEK cells when using the perforated patch-clamp technique that minimizes changes in the composition of the cytoplasm (Supplementary Fig. 1a–h). Again, mutations in either "gating-modifying" region increased current amplitudes about 10-fold and almost completely abolished gating (Supplementary Fig. 1b, c, e–h). Current amplitudes in HEK cells displayed a larger scatter probably because of a variable transfection efficiency compared to well-controlled cRNA injection into oocytes (Supplementary Fig. 1h, Fig. 1n). The difference in current amplitudes between mutations in the "gating-modifying" regions and the R172Q mutant was less pronounced than in the oocyte system (Supplementary Fig. 1d, g, h).

**Knock-in mice expressing "open ClC-2"**. With the exception of S865R that probably does not cause disease, all PA-associated *CLCN2* mutations strongly increased ClC-2 currents. We therefore predicted that any other mutation strongly enhancing Cl⁻ currents is able to cause PA. To test this hypothesis, we generated *knock-in* mice expressing an "open" ClC-2 mutant (op) that has not been found in human disease, but opens ClC-2 to the same degree as human mutations (Supplementary Fig. 2a–c). We chose to introduce a well-characterized ablation of eight N-terminal residues (YTQELGAF)[27]. This deletion "opens" ClC-2 equivalently as several other deletions in the N-terminus[27] or the J-K

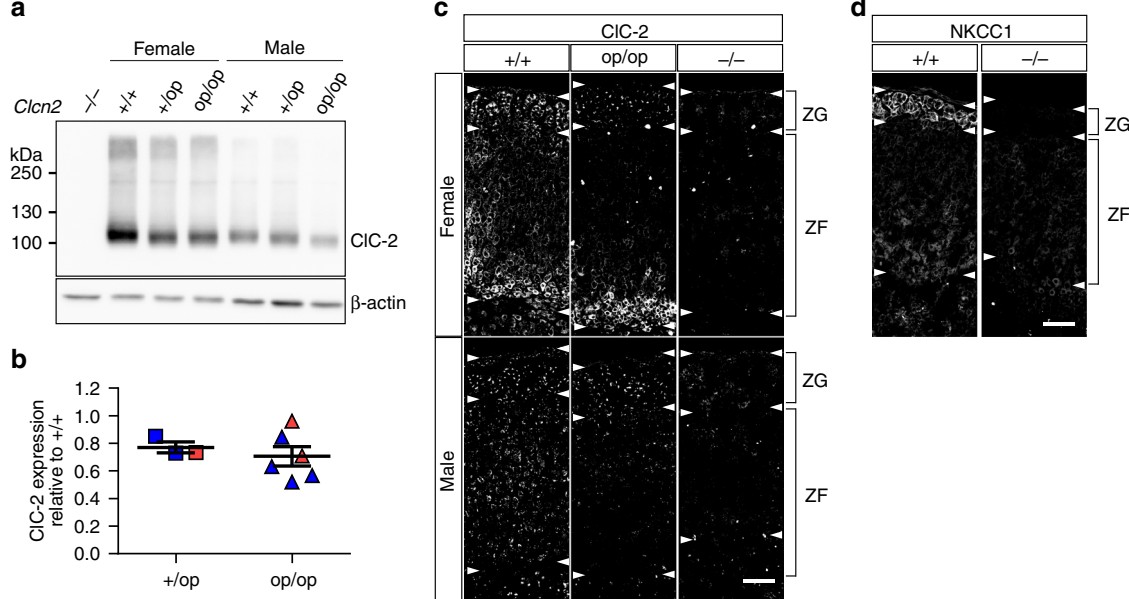

**Fig. 2** Expression of wild type and open ClC-2 protein in mouse adrenal glands. **a** Representative western blot for ClC-2 of membrane fractions isolated from adrenal gland tissue of male and female *Clcn2^+/+* (+/+), *Clcn2^+/op* (+/op), *Clcn2^op/op* (op/op) mice. Antibody specificity was verified with *Clcn2^−/−* (−/−) adrenal tissue. Equal amounts of protein were loaded, with β-actin serving as a loading control. **b** Densitometric quantification of protein bands to determine ClC-2 expression in the adrenal gland of male (blue) and female (red) *Clcn2^+/op* (squares; n = 3 animals) and *Clcn2^op/op* (triangles; n = 6 animals) mice compared to WT (n = 6 animals) assigned a reference value of 1. Short and long bars designate mean ± SEM. **c** Representative immunofluorescent staining of ClC-2 in the adrenal cortex of male and female +/+, +/op, and control −/− mice. Twelve-week-old mice were used (**a–c**). **d** Na⁺K⁺2Cl⁻ cotransporter (NKCC1) staining in the adrenal cortex of a WT (+/+) mouse, with *Nkcc1^−/−* (−/−) tissue as control (both females and 3 weeks old). **c, d** White arrowheads indicate approximate boundaries of zona glomerulosa (ZG) and zona fasciculata (ZF). Scale bar, 50 μm (**c, d**)

linker[28] and to the same extent as disease-causing human M22K, G24D, Y26N, and ΔK362 mutations (Fig. 1c–e, h, l–n, Supplementary Fig. 1b, c, e–h). Hence its effect on Cl⁻ currents is representative for most known PA-causing human mutations, with the exception of R172Q that has weaker effects (Fig. 1f, m, n, Supplementary Fig. 1d, g, h). We flanked protein-encoding exons 2 and 3 with loxP sites to allow for Cre-recombinase-mediated ablation of the gain-of-function allele in selected tissues (Supplementary Fig. 2a).

Homozygous mice (termed *Clcn2^op/op* mice, "op" for "open") were viable and lacked immediately visible phenotypes. Western blot analysis revealed expression of WT and mutant ClC-2^op in brain, cerebellum, distal colon, and adrenal gland (Supplementary Fig. 2d), with levels of mutant ClC-2^op and WT protein being comparable in most organs. To exclude that the N-terminal deletion alters the subcellular distribution of ClC-2, we examined the colon which expresses comparatively high levels of ClC-2 (Supplementary Fig. 2d). In this tissue, the channel localizes prominently to basolateral membranes of epithelial cells[39]. No differences in ClC-2 expression and localization between WT and *Clcn2^op/op* mice were detected in KO-controlled immunohistochemistry (IHC) (Supplementary Fig. 3a). This suggests that the deleted amino-terminal residues do not interact with proteins involved in ClC-2 protein trafficking and localization.

Western blots of whole adrenal glands indicated a moderate, ~20% decrease of ClC-2 levels in heterozygous *Clcn2^+/op* and homozygous *Clcn2^op/op* mice compared to WT (Fig. 2a, b). Agreeing with the wide expression pattern of ClC-2 (ref. [26]), KO-controlled IHC revealed that ClC-2 is expressed both in the ZG and ZF of the adrenal gland (Fig. 2c), contrasting with previous results using an antibody of unknown specificity[23]. In females, ClC-2 showed particularly high expression in the inner part of the ZF. Although our previous[22] and present (see below) patch-clamp analysis of ZG cells in situ showed that ClC-2 mediates plasma

membrane currents, ClC-2 expression at the plasma membrane of ZG cells was under the detection limit of IHC. Rather ClC-2 was mainly found in the trans-Golgi network (TGN) (Supplementary Fig. 3b). Likewise, colon epithelial cells, which prominently express ClC-2 at the plasma membrane, also retained a portion of ClC-2 in the TGN (Supplementary Fig. 3a). The significance of this finding is unclear.

Unlike ClC-2, the chloride-accumulating Na⁺K⁺2Cl⁻ cotransporter NKCC1 was rather specifically and strongly expressed in the ZG where it localized to plasma membranes (Fig. 2d). Hence opening of ClC-2 is predicted to result in a strongly depolarizing Cl⁻ efflux in ZG, but probably not in ZF cells.

**ZG cells expressing the *Clcn2^op* allele are depolarized**. To determine the effect of genetically opening ClC-2 on electrical parameters of ZG cells in tissue slices, we used the amphotericin-perforated patch-clamp technique. Chloride currents in WT ZG cells were barely detectable and much smaller than those we measured previously in the whole-cell mode[22] (Fig. 3a–c). ZG cells from *Clcn2^op/op* mice, however, had significantly increased Cl⁻ currents that showed little gating (Fig. 3a–c), as expected from heterologous expression in HEK cells (Supplementary Fig. 1f). In *Clcn2^+/op* mice, roughly half of the cells measured showed very little current like in WT, while the rest showed robust currents similar to *Clcn2^op/op* mice (Fig. 3a–c).

The resting potential of ZG cells was determined with gramicidin-perforated patch-clamp recordings to leave native intracellular chloride concentrations undisturbed. Agreeing with previous studies[3,40,41] and their large resting K⁺ conductance[42–44], WT ZG cells had a resting potential $V_m$ around −90 mV that is close to the potassium equilibrium potential (Fig. 3d). By contrast, cells from *Clcn2^op/op* mice were strongly depolarized to roughly −40 mV. ZG cells from *Clcn2^+/op* mice, modeling

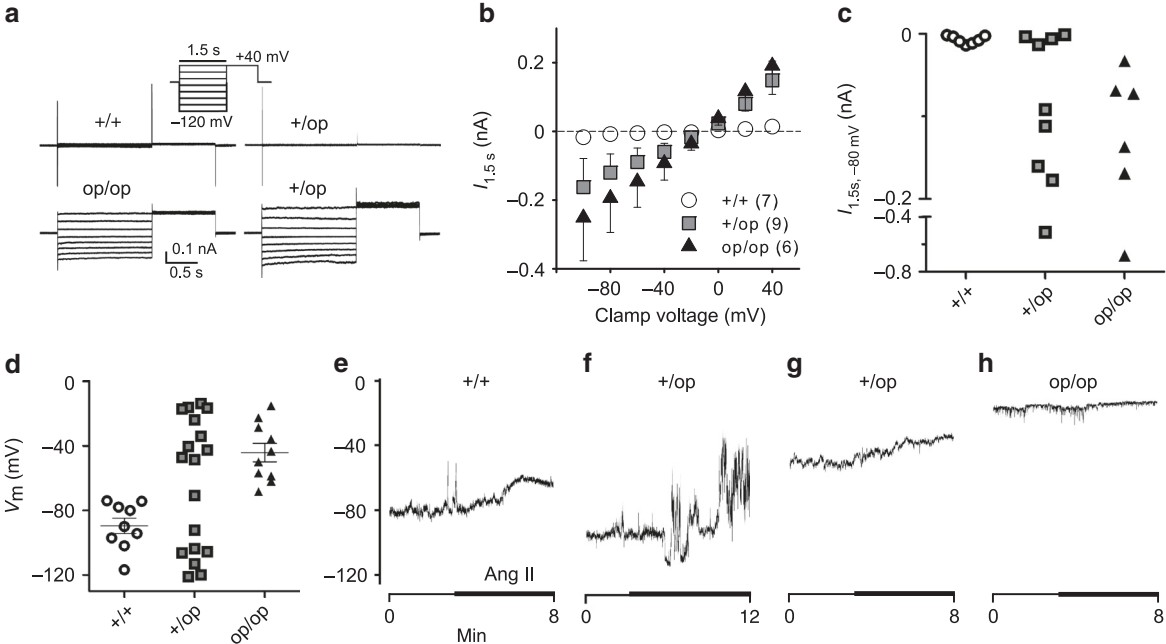

**Fig. 3** Increased chloride current and membrane depolarization in $Clcn2^{op}$ zona glomerulosa cells. **a** Representative traces of chloride currents in ZG cells from $Clcn2^{+/+}$ (7 cells, 7 mice), $Clcn2^{+/op}$ (9 cells, 8 mice), $Clcn2^{op/op}$ (6 cells from 6 mice) adrenal slices, designated as +/+, +/op, and op/op, and measured with amphotericin-perforated patch clamp using the voltage step protocol shown in inset. **b** Plot of mean ± SEM currents measured at the end of voltage steps (1.5 s) as a function of voltage from experiments performed in **a**. The number of cells measured is indicated in parenthesis. **c** Individual current values of **a**, **b** at a clamp voltage of −80 mV. **d** Individual ZG cell membrane potentials $V_m$ from $Clcn2^{op}$ mouse models using gramicidin-perforated patches with $I=0$ (9 +/+ cells from 9 mice, 18 +/op cells from 12 mice, 10 op/op cells from 8 mice) that leave intracellular chloride undisturbed. Long and short bars represent means ± SEM, respectively. Since $V_m$ distribution of +/op cells appears bimodal, no mean value is shown. **d–g** Voltage traces of +/+, +/op, op/op cells (representative of 2, 7, and 5 cells, respectively) that were superfused with 100 nM angiotensin II (Ang II) indicated by a thick line

patients who display heterozygous *CLCN2* mutations, showed a large scatter of intracellular potentials (Fig. 3d). The distribution of $V_m$ appeared bimodal, with about half of the cells displaying WT voltages and the others being strongly depolarized like in $Clcn2^{op/op}$ mice. Several cells were superfused with angiotensin II in perforated-patch recordings with current set to zero. This elicited depolarization in WT cells (Fig. 3e), but very little to none in the already strongly depolarized $Clcn2^{op/op}$ cells (Fig. 3h). Again, there were two distinct cell populations in $Clcn2^{+/op}$ mice, with one group resembling WT (Fig. 3f) and the other $Clcn2^{op/op}$ cells (Fig. 3g). Since depolarization may enhance aldosterone production by raising $[Ca^{2+}]_i$ through voltage-gated $Ca^{2+}$-channels, these findings predict that serum aldosterone in heterozygous $Clcn2^{+/op}$ mice may reach roughly 50% of the levels observed in homozygous $Clcn2^{op/op}$ mice.

**Altered calcium signaling in $Clcn2^{op/op}$ ZG cells.** $Ca^{2+}$-imaging of adrenal gland slices using ratiometric Fura-2 fluorescence revealed that basal $[Ca^{2+}]_i$ was increased in $Clcn2^{op/op}$ and $Clcn2^{+/op}$ ZG cells over WT levels (Fig. 4a–c, Supplementary Fig. 4). Whereas WT ZG cells lacked obvious spontaneous oscillations of $[Ca^{2+}]_i$ (Fig. 4d, Supplementary Video 1), 62% of ZG cells from $Clcn2^{op/op}$ mice showed marked $Ca^{2+}$-oscillations (defined by amplitudes >0.01 a.u.) that resembled those observed in $Task3^{−/−}$ mice[18] (Fig. 4f, Supplementary Video 2). Heterozygous $Clcn2^{+/op}$ mice displayed a lower proportion of cells with $Ca^{2+}$-oscillations (33% of cells), with some cells starting to show $Ca^{2+}$ transients during recordings (Fig. 4e, middle trace). $Ca^{2+}$ oscillations could be induced in WT ZG cells by angiotensin II (Fig. 4g). The increase in cytosolic $[Ca^{2+}]$ observed in $Clcn2^{op}$ mice is expected to increase the transcription of $Cyp11b2$ encoding aldosterone synthase[3].

**Aldosterone synthase transcription in $Clcn2^{op/op}$ mice.** We used quantitative reverse transcription polymerase chain reaction (qRT-PCR) to assess the expression of key enzymes of adrenal steroidogenesis, $Star$ (encoding the steroidogenic acute regulatory protein), $Cyp21a1$ (encoding steroid 21-hydroxylase, like StAR involved in both aldosterone and corticosterone synthesis pathways), $Cyp11b2$ (encoding aldosterone synthase), and $Cyp11b1$ (encoding 11β-hydroxylase responsible for the final step of corticosterone synthesis) (Fig. 5a–d). No significant differences between WT, $Clcn2^{+/op}$, and $Clcn2^{op/op}$ genotypes were observed for $Star$ and $Cyp21a1$ transcript levels (Fig. 5a, b). By contrast, significant increases in $Cyp11b2$ transcripts were observed in both male and female $Clcn2^{op/op}$ mice, with heterozygous $Clcn2^{+/op}$ mice showing a tendency for an increase (Fig. 5c). Levels of $Cyp11b1$ did not differ between genotypes (Fig. 5d), suggesting that the opening of ClC-2 channels, although expressed in both ZG and ZF (Fig. 2c), specifically stimulates aldosterone production. IHC of adrenal gland sections revealed an increased number of aldosterone synthase-positive cells in the ZG of $Clcn2^{op/op}$ mice (Fig. 5e, f). Hematoxylin-eosin staining and IHC for the ZG cell marker Dab2 (ref. [45]) did not reveal major morphological changes of adrenal glands, such as ZG area (Supplementary Figs. 5 and 6). No adenomas were detected in male or female $Clcn2^{op/op}$ mice up to an age of 52 weeks.

**Aldosterone levels in homo- and heterozygous $Clcn2^{op}$ mice.** Aldosterone levels and renin activity were determined in serum of WT, $Clcn2^{+/op}$, and $Clcn2^{op/op}$ mice of both genders (Fig. 6). Serum aldosterone was increased in $Clcn2^{op/op}$ mice by more than 10-fold and to a lesser degree in $Clcn2^{+/op}$ mice (Fig. 6a). As typical for PA, renin activity was strongly suppressed in homozygous, and to a lesser extent in heterozygous, $Clcn2^{op}$ mice

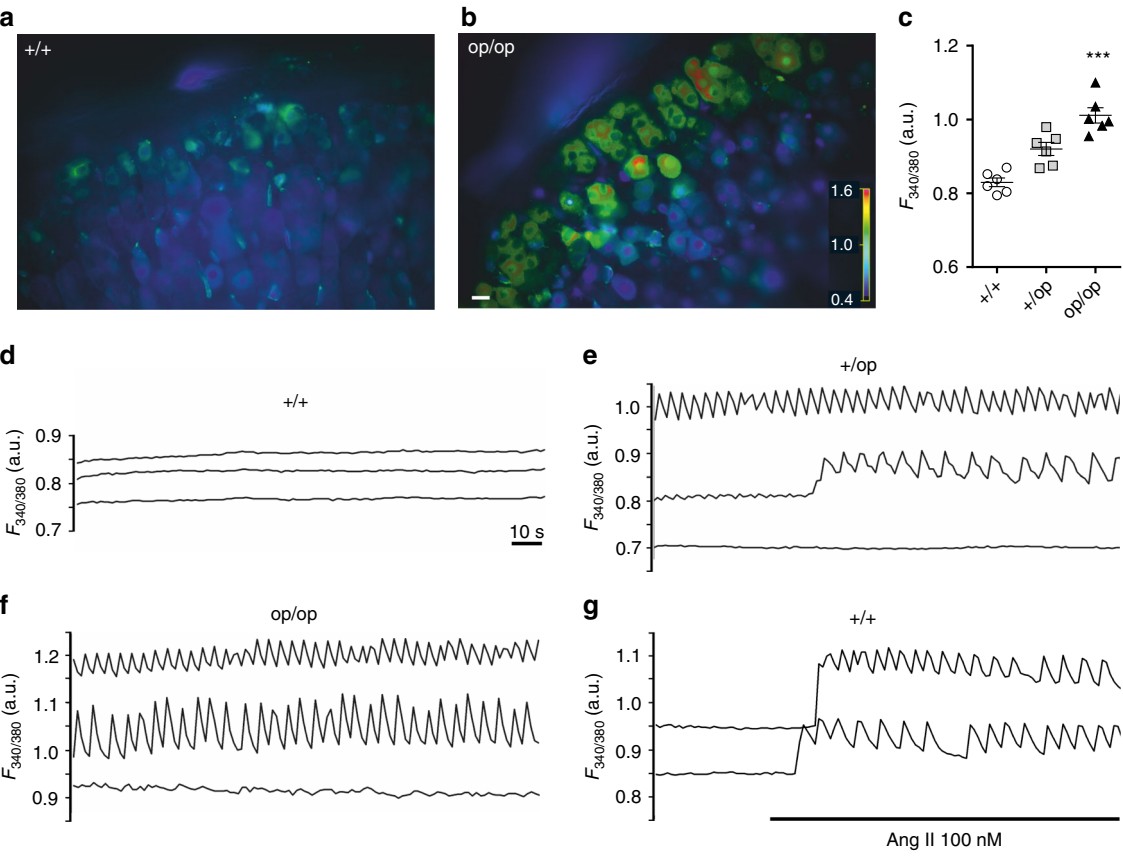

**Fig. 4** Increased intracellular calcium in *Clcn2*op zona glomerulosa cells. **a**, **b** Representative images of *Clcn2*+/+ (**a**) and *Clcn2*op/op (**b**) male adrenal Fura-2 AM-loaded slices. Colors indicate $Ca^{2+}$ fluorescence under basal conditions. Scale bar, 10 μm. **c** Quantification of $Ca^{2+}$ fluorescence from *Clcn2*+/+ (+/+, circles), *Clcn2*+/op (+/op, squares), and *Clcn2*op/op (op/op, triangles) adrenal gland slices. For each genotype, six mice were analyzed. Single values represent mean values of cells measured during 2 min from one mouse. Error bars, mean ± SEM. ***$p < 0.001$, +/op and op/op each compared to +/+, Kruskal–Wallis test with Dunn's multiple comparison test. **d**–**f** Representative *Clcn2*+/+ (**d**), *Clcn2*+/op (**e**), and *Clcn2*op/op (**f**) traces of three individual ZG cells imaged under basal conditions. **g** Traces showing the $Ca^{2+}$-response of two *Clcn2*+/+ ZG cells to angiotensin II (Ang II). Fluorescence values are expressed in arbitrary units

(Fig. 6b). Corticosterone levels were unchanged (Fig. 6c). In accord with markedly increased aldosterone levels, *Clcn2*op/op mice displayed hypokalemia (Table 1).

We additionally compared serum and urine aldosterone levels, as well as serum renin activity and corticosterone levels, between WT and *Clcn2*op/op mice under low- and high-potassium diets (Supplementary Fig. 7). As expected, aldosterone was increased with high-potassium diet in WT mice (Supplementary Fig. 7a). In *Clcn2*op/op mice, aldosterone levels were increased, and renin activity decreased, under both low- and high-potassium diets (Supplementary Fig. 7a, b). Low-potassium diet appeared to slightly blunt the increase of aldosterone levels in male *Clcn2*op/op mice, suggesting that physiological regulatory mechanisms of aldosterone production were not completely abolished. Corticosterone levels were unchanged (Supplementary Fig. 7c).

Although our data provide overwhelming evidence for PA in *Clcn2*op/op mice, the regulation of aldosterone biosynthesis involves feedback mechanisms from several tissues and ClC-2 is widely expressed[26]. We therefore wanted to ascertain that the effect of the *Clcn2*op allele on aldosterone biosynthesis is ZG cell-intrinsic. We crossed *Clcn2*op/op mice (in which exons 2 and 3 of the *Clcn2*op allele are floxed) with SF1-Cre mice that express the recombinase specifically in the adrenal cortex, somatic gonad

cells, and a few other tissues[46,47]. IHC confirmed the absence of ClC-2 from the adrenal cortex of these mice (Supplementary Fig. 8a). Whereas aldosterone levels were strongly increased in *Clcn2*op/op mice (Fig. 6a), they were undistinguishable from WT in SF1-Cre; *Clcn2*op/op mice (Supplementary Fig. 8c). Moreover, adrenal Cyp11b2 levels (Supplementary Fig. 8b) and serum renin activity (Supplementary Fig. 8d) were similar to WT. Hence, mutant ClC-2op increased aldosterone production in a ZG cell-intrinsic manner. Moreover, the loss of ClC-2 in ZG cells of SF1-Cre; *Clcn2*op/op mice did not lower aldosterone below WT levels.

**Open ClC-2 mutant increases BP.** Telemetry was used to compare systolic and diastolic BP and heart rate between WT, *Clcn2*+/op, and *Clcn2*op/op mice (Fig. 7, Supplementary Fig. 9). Under control diet, systolic BP was markedly elevated in male (by ~20 mm Hg) and to a slightly more moderate degree in female *Clcn2*op/op mice (Fig. 7a). Systolic BP of heterozygous mice was intermediate between WT and *Clcn2*op/op mice. No obvious changes in diastolic BP were detected (Fig. 7b). These measurements were repeated with the same mice exposed to diets containing high $Na^+$- or $K^+$-concentrations, or low amounts of $K^+$ (Supplementary Fig. 9). Again, systolic BP was increased in male *Clcn2*op/op

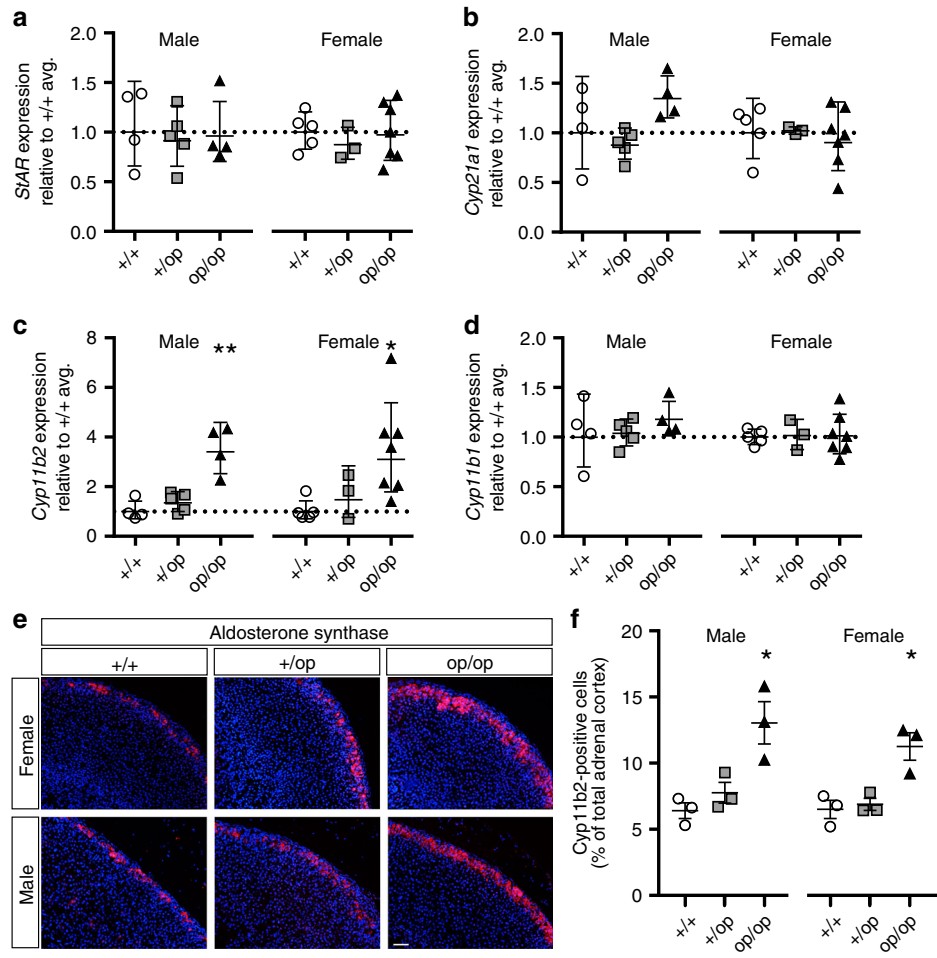

**Fig. 5** Aldosterone synthase in *Clcn2*op adrenal gland. **a–d** Quantitative RT-PCR analysis of steroidogenic enzyme mRNA expression in adrenal glands from 12-week-old male and female *Clcn2*+/+ (+/+, circles), *Clcn2*+/op (+/op, squares), and *Clcn2*op/op (op/op, triangles) mice, $n \geq 3$ (each point represents one animal). Relative expression levels (compared to *Clcn2*+/+ average, reference value of 1 indicated with a dotted line) for **a** StAR (steroidogenic acute regulatory protein), **b** Cyp21a1 (steroid 21-hydroxylase), **c** Cyp11b2 (aldosterone synthase), **d** Cyp11b1 (11β-hydroxylase). Error bars in **a–d**, geometric mean ± geometric SD. Statistical analyses was performed on ΔCt values. *$p < 0.05$, **$p < 0.01$ (+/op and op/op each compared to +/+ of same sex, male and female analyzed separately, Kruskal–Wallis test, Dunn's multiple comparison test). **e** Representative aldosterone synthase/Cyp11b2 staining of paraffin-embedded adrenal gland slices from male and female +/+, +/op, op/op mice at age 12 weeks. **f** Quantification of Cyp11b2-positive cells observed in **e** on an automated molecular imaging platform (Vectra, Perkin Elmer). $n = 3$ animals for each group. Error bars, means ± SEM. *$p < 0.05$ (+/op and op/op each compared to +/+ of same sex, male and female analyzed separately, one-way ANOVA, Bonferroni multiple comparison test). Scale bar, 100 μM

mice, with heterozygous *Clcn2*+/op appearing to have less than half of the BP increase and females showing less effect (Supplementary Fig. 9a). There were no marked changes in diastolic BP with the exception of female mice under high-Na+ diet (Supplementary Fig. 9b). Marked changes in heart rate were only observed with *Clcn2*op/op mice fed low-K+ diet (Supplementary Fig. 9c).

**Serum electrolytes and renal function in *Clcn2*op mice.** *Clcn2*op/op mice displayed hypokalemia (Table 1), a frequent abnormality in PA patients including several of those with *CLCN2* mutations[22,23]. Hypokalemia results from the stimulatory effect of aldosterone on distal tubular K+-secretion that serves to electrically compensate for increased Na+-reabsorption through aldosterone-regulated ENaC channels. Under low-K+ diet we additionally observed hypochloremic metabolic alkalosis (low serum Cl− and more alkaline pH) (Supplementary Table 1). It is probably caused by increased renal tubular H+ secretion, which is needed for the electric compensation of increased Na+ reabsorption particularly when charge compensation by K+ currents is reduced by severe hypokalemia[4]. The resulting increase in

plasma HCO3− anions is compensated by a reduction of Cl− anions (Supplementary Table 1).

In PA, high aldosterone levels, together with high BP, can lead to renal damage that may result in albuminuria[1,6,8,9,48]. Whereas we did not find obvious signs of renal damage in standard histology, *Clcn2*op/op mice displayed moderate albuminuria (Supplementary Fig. 10). It compared favorably to that of *Clcn5 knock-out* mice which display impaired proximal tubular endocytosis[49]. However, unlike *Clcn5 knock-out* mice, *Clcn2*op/op mice did not display urinary loss of vitamin D-binding protein (DBP) or retinol-binding protein (RBP), both of which are lost into the urine upon proximal tubular dysfunction[49]. Hence albuminuria of *Clcn2*op/op mice likely resulted from increased glomerular filtration. Although end organ damage was not explored further, this observation suggests that *Clcn2*op/op mice might prove useful to investigate secondary tissue damage in PA.

## Discussion
Mutations in several genes encoding ion channels of ZG cells cause human PA[10,11,50]. Progress in understanding the

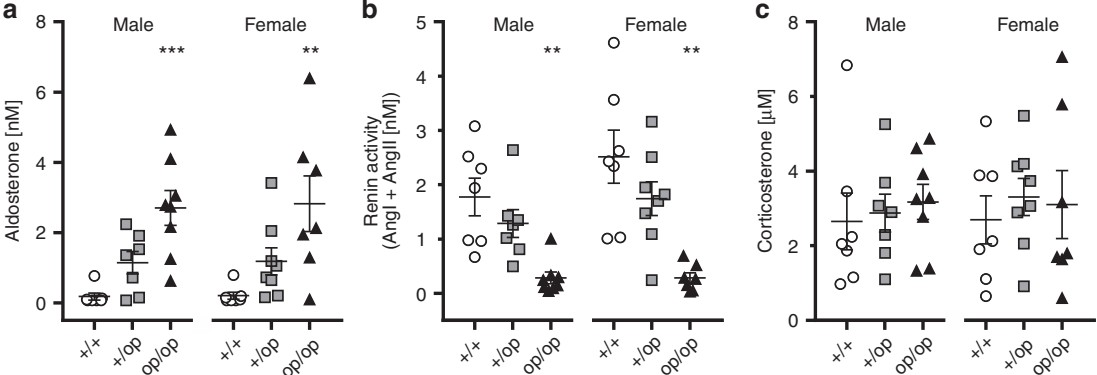

**Fig. 6** Elevated aldosterone levels and renin activity in serum of *Clcn2*op mice. **a–c** Serum levels of aldosterone (**a**), renin activity (**b**), and corticosterone concentration (**c**) of *Clcn2*+/+ (+/+, circles), *Clcn2*+/op (+/op, squares), and *Clcn2*op/op mice (op/op, triangles) aged 12 weeks and sub-grouped between males (left) and females (right) (one sample per animal was measured). Aldosterone levels below detection were set at the measureable lower limit threshold value of 70 pM (number of samples below threshold per samples measured: males: *Clcn2*+/+ $n = 4/7$, *Clcn2*+/op $n = 1/7$, *Clcn2*op/op $n = 0/8$; females: *Clcn2*+/+ $n = 3/7$, *Clcn2*+/op $n = 0/8$, *Clcn2*op/op $n = 0/7$). Renin activity was calculated as the sum of angiotensin I (Ang I) and angiotensin II (Ang II) (see Methods). Error bars, mean ± SEM. Statistical significance (Kruskal–Wallis test with Dunn's multiple comparison test; analyzed separately by sex and compared to *Clcn2*+/+) is shown above the groups: **$p < 0.01$, ***$p < 0.001$. Upon pooling both males and females in **a**, statistical analysis (Kruskal–Wallis test with Dunn's multiple comparison test) of *Clcn2*+/+ versus *Clcn2*+/op was statistically significant (**) for increased aldosterone

| **Table 1 Blood electrolyte concentrations and pH of WT, *Clcn2*+/op, and *Clcn2*op/op mice** | | | | | | | |
|---|---|---|---|---|---|---|---|
| Sex | Genotype | *n* | Na+ | K+ | Cl− | pH | HCO−3 |
| m | *Clcn2*+/+ | 8 | 149.1 ± 0.6 | 3.6 ± 0.2 | 114.4 ± 2.1 | 7.25 ± 0.04 | 22.2 ± 2.1 |
| m | *Clcn2*+/op | 12 | 150.2 ± 1.2 | 3.7 ± 1.0 | 115.3 ± 2.1 | 7.25 ± 0.05 | 20.8 ± 2.7 |
| m | *Clcn2*op/op | 11 | 150.6 ± 1.5* | 2.9 ± 0.3**** | 112.2 ± 3.0 | 7.29 ± 0.05 | 22.9 ± 2.4 |
| f | *Clcn2*+/+ | 8 | 149.1 ± 1.0 | 3.4 ± 0.3 | 114.1 ± 2.5 | 7.27 ± 0.04 | 20.9 ± 1.8 |
| f | *Clcn2*+/op | 11 | 149.5 ± 2.2 | 3.0 ± 0.3 | 114.7 ± 2.7 | 7.27 ± 0.04 | 21.9 ± 1.5 |
| f | *Clcn2*op/op | 10 | 150.4 ± 1.8 | 2.6 ± 0.2*** | 114.3 ± 1.6 | 7.27 ± 0.04 | 21.4 ± 1.6 |

*m* male, *f* female. Ion concentrations in mM. *n*, number of mice. Significant difference with reference to *Clcn2*+/+: *$p < 0.05$, ***$p < 0.001$, **** $p < 0.0001$; Mean values ± SD; Kruskal–Wallis test, Dunn's multiple comparison test

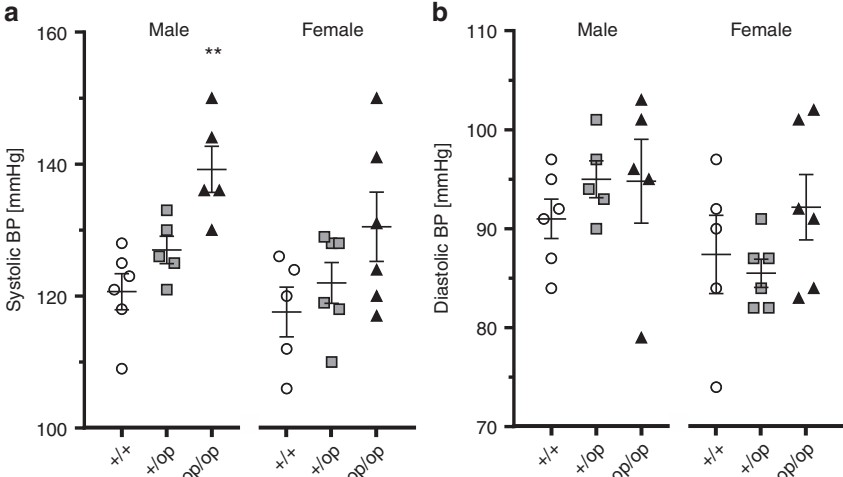

**Fig. 7** Elevated blood pressure in *Clcn2*op mice. **a**, **b** Systolic (**a**) and diastolic (**b**) blood pressure (BP) measured by telemetry in *Clcn2*+/+ (+/+, circles, males: $n = 6$, females: $n = 5$), *Clcn2*+/op (+/op, squares, $n = 5$ of each sex) and *Clcn2*op/op mice (op/op, triangles, males: $n = 5$, females: $n = 6$) fed with control diet and sub-grouped by sex (males, left; females, right). Single values represent the mean blood pressure of an individual mouse measured over 7 consecutive days. Error bars, mean ± SEM. Statistical significance (analyzed separately by sex and compared to *Clcn2*+/+) is shown above the groups: **$p < 0.01$ (Kruskal–Wallis test, Dunn's multiple comparison test)

underlying pathological mechanisms has been hampered by the absence of mouse models that faithfully replicate the human disease[51]. Here we describe mouse models for the recently identified *CLCN2*-related form of human PA[22,23]. These mice display high aldosterone and low renin levels, marked hypertension, hypokalemia, and signs of incipient kidney damage. *Clcn2*[op] mice constitute the best mouse model for PA available to date. The *Clcn2*[op] allele, which encodes, like other PA-causing *CLCN2* mutants, an "open" plasma membrane Cl⁻ channel, increased aldosterone production by ZG cells in a cell-autonomous manner. Opening of ClC-2 strongly depolarized native ZG cells, increased $[Ca^{2+}]_i$, and stimulated transcription of aldosterone synthase-encoding *Cyp11b2*. Surprisingly, in heterozygous *Clcn2*[+/op] mice roughly half of the ZG cells were as strongly depolarized as in homozygous *Clcn2*[op/op] mice, whereas the other half displayed WT voltages. Consequently, the increases in aldosterone levels and BP were less pronounced in heterozygous than in homozygous *Clcn2*[op] mice. The effect of the artificial "opening" mutation in our *Clcn2*[op] mice strongly suggests that also the human PA-causing *CLCN2* mutations, all of which open ClC-2, cause disease by their increased chloride currents rather than by other mechanisms. It further suggests that mutations increasing other Cl⁻ currents in ZG cells may also provoke aldosteronism.

Consistent with our analysis of the PA-associated mutant p. Gly24Asp[22] we now found that most missense mutants described by others[23] also markedly increased ClC-2 currents. The only exception is S865R, which was found in a single patient and which likely is a silent polymorphism. The largest effects were observed with mutations in previously identified "inactivation" or "gating-modulating" domains in the cytosolic N-terminus[27] and the J-K linker[28]. In our previous structure–function studies we found that the resulting "open" mutants displayed >10-fold increased current amplitudes and had lost voltage-dependent gating. Those "open" mutants that were tested had also lost their activation by cell swelling or moderately acidic pH[27,28]. Some mutants displayed an "intermediate" phenotype as defined by more moderately increased current amplitudes, reduced, but not abolished gating, and intact pH dependence. Whereas the mutants in the "gating-modulating" regions (M22K, G24D, Y26N, ΔK362) resulted in "open" ClC-2 channels, the R172Q mutant rather showed an "intermediate" effect characterized by less strongly increased current amplitudes, persistent voltage-dependent gating, and pH dependence[27,28]. Intriguingly, only this mutation was found in several families. It led to dominantly inherited hyperaldosteronism in three generations of the same family, but was also found in only mildly affected or even unaffected individuals[23]. The more frequent observation of this less severe mutation might result from an increased probability of inheritance.

The mechanism by which the two previously identified regions[27,28] modulate gating remains obscure. The gating of an "open" N-terminal deletion mutant could be rescued by transplanting the deleted segment into the ClC-2 C-terminus[27]. This position-independence suggests that this segment binds either to the channel, with the J-K loop being a candidate binding site, or to an associated protein and thereby partially closes the channel[27,28]. When this binding is lost or weakened with deletions or point mutations, channels show an "open" phenotype. The situation is even more complex. It later emerged that the "opening" effect of N-terminal mutants was largely lost in whole-cell or excised-patch experiments in which the mutants only displayed moderately changed gating[25,37,38]. The mutational "opening" of ClC-2 thus required an almost intact cytoplasm as preserved in TEVC of *Xenopus* oocytes or in perforated-patch-clamp measurements of transfected mammalian cells. This

suggests that a so-far unknown, diffusible cytoplasmic factor is necessary for the full manifestation of the "open" phenotype. By using TEVC of oocytes and perforated-patch recordings of HEK cells we now revealed the full potential of the mutations described by Scholl et al.[23]. The observation that WT ClC-2 currents could be detected in previous whole-cell measurements[22], but not with perforated-patch recordings of ZG cells in situ (Fig. 3a), and that they were neither observed in perforated-patch recordings of WT ClC-2 transfected H295R-S2 cells[22], also points to the importance of an intact cell interior.

The depolarization of ZG cells is obviously not linearly related to the magnitude of an additional Cl⁻ conductance (such as that mediated by ClC-2[op]) because $V_m$ can maximally reach the Cl⁻ equilibrium potential. Therefore, even if *Clcn2*[+/op] cells express only half the amount of open ClC-2[op] channels compared to *Clcn2*[op/op] cells, they might be depolarized to (almost) the same degree as *Clcn2*[op/op] cells if their Cl⁻ conductance suffices to drive $V_m$ to near-saturation. This would result in a fully dominant effect of the *Clcn2*[op] allele with respect to aldosterone secretion, hypertension, and other parameters. This was clearly not the case in *Clcn2*[+/op] mice. This observation could not be explained by an intermediate degree of depolarization of *Clcn2*[+/op] cells. Surprisingly, about half of *Clcn2*[+/op] cells were as strongly depolarized as *Clcn2*[op/op] cells (compatible with the notion that expression from one *Clcn2*[op] allele suffices for maximal depolarization), whereas the other cells remained hyperpolarized (Fig. 3d). This finding explains the less pronounced phenotype of heterozygous mice in a straightforward way. However, the reason for the apparently bimodal distribution of $V_m$ of *Clcn2*[+/op] ZG cells remains unclear, in particular since ClC-2 appears homogeneously expressed in the adrenal cortex. In principle, a bimodal distribution is compatible with monoallelic expression of *Clcn2*, with ~50 % of cells expressing the *Clcn2*[op] and the remainder the WT allele. Monoallelic expression is not rare in mammals[52–55] and bioinformatics suggests that *Clcn2* might be expressed monoallelically in some tissues[56]. We examined this possibility by staining tissue sections of *Clcn2*[+/−] mice for ClC-2. With strict monoallelic *Clcn2* expression a mosaic-like ClC-2 expression pattern would be expected. However, we failed to find an obvious difference between *Clcn2*[+/−] and WT mice in ClC-2 IHC of adrenal cortex. We repeated these experiments with colon which expresses higher levels of ClC-2. Again, no evidence for a mosaic ClC-2 expression in *Clcn2*[+/−] mice was obtained (Supplementary Fig. 3a). However, we cannot rule out that ClC-2 is expressed at any given time only predominantly from one allele, in particular since transcription can occur in stochastic bursts from either allele[54,55]. In this scenario, one could imagine that, due to sensitivity and saturation issues, IHC might not strictly correlate with the more quantitative and sensitive electrophysiology. Alternatively, one may hypothesize that, as a result from a nonlinear interplay of various ion conductances and intracellular ion concentration, $V_m$ of ZG cells has two metastable states in the presence of a medium-sized depolarizing current (such as potentially expressed from one *Clcn2*[op] allele). However, the fact that also ClC-2 currents of ZG cells appeared bimodally distributed (Fig. 3c) rather points to an upstream factor controlling the amplitude of these currents. In this respect, it is important to point out that WT ZG cells, as identified by morphology and Dab2 staining[45], express aldosterone synthase heterogeneously[18,45,57–59] (Fig. 5e, Supplementary Fig. 6). The mechanism for this expression pattern remains enigmatic and it is unclear whether it relates to the bimodal distribution of $V_m$ in *Clcn2*[+/op] mice.

In any case, ClC-2 currents are unlikely to influence aldosterone secretion in WT animals since they are small compared to the resting K⁺ conductance of ZG cells[22] which drives $V_m$ close to

the K$^+$ equilibrium potential. This notion is supported by SF1-Cre; *Clcn2*$^{op/op}$ mice which lack ClC-2 in ZG cells but display normal aldosterone.

*Clcn2*$^{op/op}$ ZG cells depolarized to an average of roughly −40 mV and even reached voltages around −20 mV. If this depolarization is directly caused by a Cl$^-$ conductance, intracellular Cl$^-$ concentrations ([Cl$^-$]$_i$) must be at least 30 and 70 mM, respectively, values which can be reached when cells prominently express a Cl$^-$ loader such as NKCC1 (as shown here). A median value for [Cl$^-$]$_i$ ~75 mM has indeed been reported for ZG cells[23]. The observed depolarization is amply sufficient to open L- and T-type Ca$^{2+}$-channels which we had implicated in *Clcn2*$^{G24D}$-dependent aldosterone secretion in pharmacological experiments[22]. The resulting increase in [Ca$^{2+}$]$_i$, as ascertained here, is a strong trigger for the transcription of aldosterone synthase (CYP11B2)[3].

Although ClC-2 is expressed both in ZG and fasciculata cells, *Clcn2*$^{op}$ mice displayed high serum aldosterone, but no increase in corticosterone levels. Likewise, the patient with the p.Gly24Asp mutation had normal cortisol levels[22]. Since the Cl$^-$ loader NKCC1 appears to be expressed in ZG, but not in ZF cells (Fig. 2d), only ZG cells may display the high intracellular Cl$^-$ concentration required for a depolarizing Cl$^-$ efflux and the subsequent increase of [Ca$^{2+}$]$_i$. Indeed, measurements with Cl$^-$-sensitive dyes revealed high [Cl$^-$]$_i$ only in the ZG[23]. Moreover, unlike aldosterone synthesis, stimulation of corticosterone production depends mainly on cAMP rather than on [Ca$^{2+}$]$_i$[2].

While these considerations explain the specific effect of open ClC-2 channels on aldosterone versus corticosterone secretion, the wide expression pattern of ClC-2 (ref. [26]) suggests that there may be abnormalities in other cells and tissues of *Clcn2*$^{op}$ mice. Though no consistent additional symptoms were reported in patients with *CLCN2*-related PA[22,23], and although *Clcn2*$^{op/op}$ mice appeared grossly normal, it seems worthwhile to investigate possible effects of constitutively increased plasma membrane Cl$^-$ conductance in other organs. It is important to recall that the opening of ClC-2 may have divergent effects in different tissues because [Cl$^-$]$_i$ varies widely between cell types. In many cells, like adrenal ZG cells and most epithelial cells, [Cl$^-$]$_i$ is above electrochemical equilibrium. Under these conditions, Cl$^-$ channel opening causes a depolarization and possibly a rise in [Ca$^{2+}$]$_i$ as found here. In most neurons, however, [Cl$^-$]$_i$ is low due to the presence of the K$^+$Cl$^-$-cotransporter KCC2 (ref. [60]) and opening of Cl$^-$ channels entails hyperpolarization that may blunt neuronal excitability. We anticipate to identify additional symptoms in *Clcn2*$^{op/op}$ mice that may not only be relevant for basic science but also for clinical medicine. In any case, *Clcn2*$^{op}$ mice provide an excellent mouse model for human PA that will prove useful to explore its pathophysiology, and possibly therapeutic approaches, in detail.

## Methods

**Mice**. All animal experiments were approved by the relevant local authorities (LAGeSo Berlin, Germany) and performed in compliance with the relevant regulations. If not otherwise stated, all mice were housed under standard conditions (chow: 0.24% sodium, 0.49% chloride, 0.98% potassium) in the MDC animal facility according to institutional guidelines. Experiments were performed at different ages (indicated in figure legends) and with both sexes, which were analyzed separately. *Clcn2*$^{-/-}$ mice[31] and *Nkcc1*$^{-/-}$ mice[61] have been described previously.

To generate *Clcn2*$^{op/op}$ mice, a 10.6-kb fragment of R1-ES cell genomic DNA containing exons 1–22 of *Clcn2* was cloned into pKO Scrambler Plasmid 901 (Lexicon Genetics Inc.). A neomycin-resistance cassette flanked by FRT sites was inserted between exons 1 and 2. A deletion of 24 bp (TACACTCAGGAACTCG GGGCCTTT; aa sequence: YTQELGAF) followed by an additional change of two amino acids (AK to IS; bp change: GCCAAA to ATATCA) to create an *Eco*RV site to aid Southern blot analyses were inserted in exon 2. Additionally two loxP sites were introduced flanking exons 2 and 3. The linearized vector was electroporated into R1 ES cells and neomycin-resistant clones were screened by Southern blotting. Correctly targeted clones were injected into C57BL/6 blastocysts by the MDC

transgenic facility and the resulting chimeric animals were crossed to *FLP*-deleter mice[62] (in C57BL/6 background) to remove the neomycin-resistance cassette. For PCR genotyping primers 5′-GAGCGAGGGGTGGGGTCAAAACCC-3′ (P1) and 5′-CTGGATCTTACTGCGACATCTGGC-3′ (P2) were used to detect the deletion. *Clcn2*$^{op/op}$ mice were in a C57BL/6-129/Svj mixed genetic background. Additionally, we crossed *Clcn2*$^{op/op}$ mice (in which exons 2 and 3 of the *Clcn2*$^{op}$ allele are floxed) with SF1-Cre mice[46] (The Jackson Laboratory; Tg(Nr5a1-cre) 7Lowl/J) that express the recombinase specifically in the adrenal cortex, somatic gonad cells, pituitary, and SF1-positive neurons in the ventromedial hypothalamic nucleus.

**Antibodies**. The rabbit anti-mouse ClC-2 antibody (1:500 for IHC and western blot (WB)) against a C-terminal peptide of mouse ClC-2 ((CWGPRSRHGLPR EGTPSDSDDKSQ) has been described[30]. Rabbit anti-mouse NKCC1 antibody (1:250 for IHC) was raised against peptide CDEEEDGKTPTQPLL (coupled to keyhole limpet hemocyanin via the N-terminal cysteine). It was affinity-purified against the peptide and KO controls proved its specificity in several tissues. The following commercial antibodies were used: sheep anti-TGN38 (AHP499, Bio-Rad, 1:200 for IHC), mouse anti β-actin (clone AC-74, Sigma, A2228, 1:1,000 for WB), sheep anti albumin (Biotrend, 1:5000 for WB), goat anti transferrin (Immunovision, 1:250 for WB), rabbit anti vitamin D-binding protein (DBP, Bio-Rad, 9580-2710, 1:300 for WB), rabbit anti Dab2 (1:100 for IHC, sc-13982, Santa Cruz Biotechnology). Rabbit anti Cyp11b2 (1:25 for IHC) was a generous gift from CE Gomez-Sanchez (University of Mississippi Medical Center, USA). Rabbit anti-retinol binding protein (RBP) was a generous gift from Bill Blaner (Columbia University, NY, USA).

**Western blot analyses**. For WB analyses of ClC-2 protein in mouse tissue, membrane fractions were isolated from frozen tissue[28]. Membrane-containing pellets were resuspended by sonication in 50 mM Tris pH 6.8, 140 mM NaCl, 0.5 mM EDTA, and 1% SDS and 1% Triton X-100 with protease inhibitors (4 mM Pefabloc and Complete® EDTA-free protease inhibitor cocktail, Roche). Equal amounts (10–20 μg) of protein were separated by SDS-PAGE and blotted onto nitrocellulose. ClC-2 protein bands were quantified using Image Studio Lite Ver 5.2. To collect 24-h urine, mice were housed in metabolic cages (see below). Urine samples were adjusted to creatinine content and analyzed by western blot for albumin, transferrin, vitamin D-binding protein (DBP), and retinol-binding protein (RBP).

**Immunohistochemistry**. Immunohistochemical staining (IHC) for ClC-2, TGN38, and NKCC1 was performed on cryosections from adrenal glands of non-perfused 3–12-week-old mice fixed in 1% paraformaldehyde (PFA)/PBS for 1 h at 4 °C, or from distal colon of mice perfused with 1% PFA/PBS under deep anesthesia. Tissue was equilibrated in 30% sucrose at 4 °C overnight and cryo-embedded in Tissue Tek OCT compound (Sakura Finetek). Six micrometer cryosections were prepared and postfixed with 1% PFA/PBS for 10 min, blocked and permeabilized for 30 min in 0.25% Triton X-100/5% normal donkey serum (NDS)/PBS. Antibodies were diluted in 5% NDS/0.1% Tween-20/PBS. Incubation with primary antibody was performed at 4 °C overnight, and with secondary antibodies (donkey anti-rabbit Alexa Fluor 488, 1:1000, Thermo Fischer Scientific and donkey anti-sheep Northern Lights 557, 1:500, R&D Systems) for 1 h at room temperature (RT). Images were acquired using an LSM880 confocal microscope and ZEN software and are representative of $n \geq 2$ animals analyzed per genotype for each sex.

IHC of Cyp11b2 (aldosterone synthase) and Dab2 protein was performed on paraffin sections of adrenal glands from 12-week-old mice. Sections were permeabilized with TBS/0.1% Triton X-100 for 15 min and blocked with 5% BSA/ TBS for Dab2 or with 10% NDS/0.1% Triton X-100/TBS for Cyp11b2. Primary antibodies were diluted in 3% BSA/TBS for Dab2 or 5% NDS/10% normal rat serum/0.1% Triton X-100/TBS for Cyp11b2 and incubated for 1 h at RT (Dab2) or overnight at 4 °C (for Cyp11b2). Primary antibodies were detected with donkey anti-rabbit Alexa 594 (1/400 for Dab2, 1/500 for mCyp11b2, A21207, Thermo Fisher). Nuclei were counterstained using 4′,6-diamidino-2-phenylindole (DAPI) (1/5000, Roche Diagnostics GmbH). Multispectral images were acquired using a Vectra® automated imaging system and automatically quantified with InForm® software (both Perkin Elmer). Results of quantification are expressed in percentage of Cyp11b2- or Dab2-positive ZG cells over total adrenal cortex cells. Statistical significance was assessed among groups using one-way ANOVA test followed by Bonferroni's multiple comparison test.

**Histology**. To collect adrenal glands for histological analyses, mice were perfused with 1% PFA/PBS under deep anesthesia or alternatively sacrificed via cervical dislocation and the organs were postfixed overnight in 1% PFA/PBS. Haematoxylin–eosin staining was performed with 6 μm sections of paraffin-embedded organs. ZG area was quantified relative to the whole adrenal gland area using ImageJ software.

**qRT-PCR**. For qRT-PCR total RNA from the right adrenal gland of 12-week-old mice was extracted using the NucleoSpin RNA XS kit (Macherey-Nagel) according to the manufacturer's protocol. Frozen tissue was homogenized with a T10 basic

ultra-turrax (IKA). cDNA was synthesized by reverse transcription from 500 ng of total RNA using SuperScript II Reverse Transcriptase and random primers (Invitrogen). qPCR reactions were prepared using 300 nM primers and Power SYBR Green PCR Master Mix (Applied Biosystems) and run on a StepOnePlus Real-Time PCR System (Applied Biosystems). Relative mRNA expression levels of steroidogenic enzymes (compared to average of $Clcn2^{+/+}$ animals of same sex) were calculated using the $\Delta\Delta C_T$ method. Normalization involved three reference genes (mean of *Ubiquitin C*, *ß-actin*, and *Gapdh* expression). Statistical analyses was performed at the $\Delta C_T$ stage. The following primers were used:

5′-AGCCCAGTGTTACCACCAAG-3′ and 5′-ACCCAAGAACAAGCACA AG-3′ for *Ubiquitin C*;

5′-TGTGATGGTGGGAATGGGTCAGAA-3′ and 5′-TGTGGTGCCAGA TCTTCTCCATGT-3′ for *ß-actin*;

5′-AGCCTCGTCCCGTAGACAAAA-3′ and 5′-TGGCAACAATCTCCA CTTTGC-3′ for *Gapdh*;

5′-TGTATCGAGAGCTGGCAGAG-3′ and 5′-CCTGGATGGCATCCA TTGAC-3′ for *Cyp11b1*;

5′-ACCTACAGTGGCATTGTG-3′ and 5′-GATTGCTGTCGTGTCAAC-3′ for *Cyp11b2*;

5′-GTGCTTCATCCACTGGCTGGAA-3′ and 5′-GTCTGCGATAGGACCTG GTTGA-3′ for *Star*;

5′-CTTCACGACTGTGTCCAGGACT-3′ and 5′-CCGACTCTCTTGGATCTG CTTC-3′ for *Cyp21a1*.

**Electrophysiology**. TEVC measurements of *Xenopus* oocytes were performed[22]. Mutant channel cDNAs were generated by site-directed mutagenesis and cRNAs were prepared from pFROG vectors using the mMESSAGE mMACHINE T7 kit (Ambion) and injected into *X*. oocytes, at 14 ng per cell for WT ClC-2 (human, rat) and 9 ng per cell for mutants (human: M22K, G24D, Y26N, R172Q, ΔK362, S865R; rat: G30D, op). Following 2 days of expression at 17 °C, TEVC was performed at RT using a TurboTEC amplifier (npi electronic) and pClamp software (Molecular Devices) to elicit ClC-2 currents (2 s steps from +60 to −120 mV with a final 1 s step at +40 mV) in ND109 solution containing (in mM) 109 NaCl, 2 KCl, 1 MgCl₂, 1.8 CaCl₂, and 2 HEPES and adjusted to pH 7.4. To determine the pH sensitivity of R172Q, currents were first measured in ND109 and again following solution exchange with ND109 buffered with 5 mM Tris (pH 8.5).

Perforated-patch-clamp analysis of ClC-2 proteins were performed in HEK cells. Cells were transfected (X-fect) with bicistronic plasmids encoding emGFP (to identify transfected cells) and, after an IRES sequence, either human WT or mutant ClC-2 cDNA. Using a Multiclamp 700B amplifier (Axon Instruments), cells were measured 12–24 h later with amphotericin B-perforated patch clamp. The tips of patch pipettes were first filled with amphotericin B-free internal pipette solution, containing (in mM) 110 Cs-methanesulfonate, 30 CsCl, 4 NaCl, 10 HEPES, 1 MgCl₂, 1 EGTA (pH 7.4; 290 mOsm/L), and then backfilled with the same solution containing 200 μg/ml amphotericin B. GFP-expressing cells were selected for analysis. When adequate access resistance was attained (<30 MΩ) a bath solution containing (in mM) 140 CsCl, 10 HEPES, 1.8 MgCl₂, 1.8 CaCl₂, and 20 sucrose (pH 7.4; 300 mOsm/L) was perfused to measure anion currents in the voltage-clamp configuration (1.5 s 20 mV steps from +40 mV to either −120 mV or −100 mV with a final 1s step at +40mV). Signals were digitized at 10 kHz and filtered at 2 kHz. Measurements were performed at RT (22–24 °C).

For patch-clamp analysis of ZG cells, adrenal slices from 3- to 7-week-old mice of both sexes were prepared[22]. Bicarbonate-based bath solutions were used and continuously bubbled with 95% O₂ and 5% CO₂. Briefly, adrenal glands were removed and placed into cold low-Ca²⁺ solution containing (in mM) 140 NaCl, 2 KCl, 26 NaHCO₃, 10 glucose, 5 MgCl₂, and 0.1 CaCl₂, pH 7.4. After removal of surrounding fat tissue, adrenal glands were embedded in 3% low-melting agarose, sectioned at 70 μm (Leica, VT1200S), and incubated at RT in physiological solution containing (in mM) 140 NaCl, 2 KCl, 26 NaHCO₃, 10 glucose, 2 MgCl₂, and 2 CaCl₂ and adjusted to pH 7.4. After ≥1 h, slices were transferred to a recording chamber and imaged with a ×60 objective and DIC optics (Olympus, BX51WI). Sub-capsular cells were selected for analysis by amphotericin-perforated patch clamping. Patch pipettes were prepared as described above for analysis of transfected cells. When the membrane resistance of the perforated cell was stable, the bath solution was exchanged for continuously 95% O₂ and 5% CO₂-bubbled Na⁺, K⁺ free solution composed of (in mM) 117 NMDG-Cl, 23 NMDG-HCO₃, 5 CsCl, 1.3 MgCl₂, 9 glucose, and 2 CaCl₂, adjusted to pH 7.3. Currents were elicited using voltage steps from +40 to −100 mV from a holding potential of −10 mV, with a final 1 s step at +40 mV. For measuring membrane potentials, gramicidin-perforated patch clamp was used[22]. Briefly, the tips of patch pipettes were first filled with solution containing (in mM) 100 KMeSO₃, 30 KCl, 4 NaCl, 10 HEPES, 1 MgCl₂, 1 EGTA, and 3 MgATP (pH 7.3; 280 mOsm/L), and then backfilled with the same solution containing 25 μg/ml gramicidin. In all, ~20 min following tight giga-seal formation, stable membrane potential measurements ($I = 0$ configuration) could be measured. Some cells were superfused with angiotensin II (100nM).

**Intracellular Ca²⁺ measurements**. Left adrenal glands from male mice of age 8 to 15 weeks were placed into ice-cold cutting solution containing (in mM) 26 NaHCO₃, 125 NaCl, 1.25 NaH₂PO₄, 10 glucose, 10 HEPES, 3 MgCl₂, 0.1 CaCl₂, 2.5

KCl. After removal of surrounding fat tissue, adrenal glands were embedded in 3% low-melting agarose, cut into 250-μm-thick slices using a vibratome (VT 1200 S, Leica), and loaded with 5 μM Fura-2 AM in the presence of 1× PowerLoad™ (both Invitrogen) at 37 °C for 30 min. Imaging was performed in recording solution containing (in mM) 26 NaHCO₃, 125 NaCl, 1.25 NaH₂PO₄, 10 glucose, 10 HEPES, 1 MgCl₂, 2 CaCl₂, 2.5 KCl. Both solutions were continuously bubbled with 95% O₂ and 5% CO₂ and pH was adjusted to 7.4. Slices were alternately excited by 340 and 380 nm light using a monochromator (Polychrome V, TILL Photonics). Emission signals were recorded at 510 nm and images obtained at one frame per second with a CMOS camera (ORCA-spark, Hamamatsu) on an upright microscope (Axio Examiner.A1, Zeiss) with a water immersion ×40 objective (W N-Achroplan, Zeiss) using MetaFluor software (Molecular Devices). Intracellular Ca²⁺ is represented as the change in ratio of 340/380 fluorescent signal activity. At the end, some slices were superfused with angiotensin II (100 nM). For each individual mouse, the average Ca²⁺ fluorescence and the percentage of oscillating cells (defined by amplitudes >0.01 a.u.) were calculated.

**Radiotelemetric BP measurement**. For these experiments, mice were single housed. Adult mice (14- to 18-week-old) were anesthetized by inhalation of isoflurane (2–2.5 vol%). The radiotelemeters (PA-C40; Data Sciences International) were implanted subcutaneously, with the sensing tip placed in the aorta via the right carotid artery. After 10 days of recovery from surgery during which mice were fed a high-sodium diet (3.23% sodium, 4.90% chloride, 0.97% potassium), BP and heart rate were recorded for seven consecutive days under the same diet. Subsequently mice were kept under three other special diets (low potassium (0.21% sodium, 0.29% chloride, <0.05% potassium), high potassium (0.21% sodium, 0.29% chloride, 3% potassium), and control (0.21% sodium, 0.49% chloride, 0.97% potassium)) for 10 days for acclimatization followed by seven recording days.

**Blood and urine analysis**. Mice fed with standard diet or special diets (see above) were anesthetized with 70–150 mg/kg pentobarbital injected intraperitoneally. Seventy-five microliters of blood was collected via the retro bulbar technique for analyses of, i.e., electrolytes and pH (iSTAT handheld blood analyzer with EC8+ cartridges, Abbott Laboratories). For hormone measurements blood was collected from the beating heart of anesthetized mice. After 10–15 min at RT, the blood was centrifuged at 1600 *g* and the serum was collected for analysis.

Aldosterone and angiotensin (1–10) quantification was done by Attoquant Diagnostics GmbH (Vienna, Austria) using a liquid chromatography-mass spectrometry/mass spectroscopy (LC-MS/MS)-based approach. Murine serum conditioning for equilibrium analysis was performed at 37 °C followed by stabilization by addition of an enzyme inhibitor cocktail as described[63,64]. Previous results have shown similar qualitative outcomes when comparing the quantification of circulating and equilibrium angiotensin peptide levels[63]. Stabilized equilibrated serum samples and urine samples (see below) were further spiked with stable isotope-labeled internal standards for each angiotensin metabolite, angiotensin-(1–10) (Ang I), angiotensin-(1–8) (Ang II), angiotensin-(2–8) (Ang III), and angiotensin-(3–8) (Ang IV) as well as with deuterated internal standards for both steroids, aldosterone-D4 and corticosterone-D4, at a concentration of 200 pg/ml. Following C18-based solid-phase-extraction, samples were analyzed by liquid chromatography-mass spectrometry/mass spectroscopy (LC-MS/MS) using a reversed-phase analytical column operating in line with a Xevo TQ-S triple quadruple mass spectrometer (Waters). Internal standards were used to correct for peptide and steroid recovery of the sample preparation procedure for each angiotensin metabolite and steroid in each individual sample. Analyte concentrations were calculated considering the corresponding response factors determined in appropriate calibration curves in original sample matrix, on condition that integrated signals exceeded a signal-to-noise ratio of 10. Renin activity was calculated as the sum of Ang I and Ang II[63,64].

Mice were acclimatized for 10 days to each of the special diets (one group of mice with control and subsequently with low K⁺, and another group of mice with high K⁺; see above). To collect 24 h urine, mice were housed in metabolic cages (metabolic cage for single mouse; UNO B.V., Marconistraat 31, 6902 PC Zevenaar, Netherlands) under the respective special diet. Feces and chow were removed from the urine. Aldosterone and renin activity was determined by Attoquant Diagnostics GmbH. Urine was diluted in an Ang I generation buffer containing ethylenediaminetetraacetic acid (5 μM, Sigma-Aldrich), Z-Pro-Prolinal (20 μM, Sigma-Aldrich), 4-(2-Aminoethyl)benzenesulfonyl fluoride hydrochloride (1mM, Sigma-Aldrich), aminopeptidase inhibitor (10 μM, Sigma-Aldrich), and the substrate murine Angiotensinogen (50 μg/ml, Sino Biological) in phosphate-buffered saline (Dulbecco's PBS, pH 7.4, Sigma-Aldrich). Following incubation at 37 °C, the mixture was stabilized and further processed the same as all serum samples for quantification of angiotensin-(1–10) (Ang I) and aldosterone in LC-MS/MS analysis. For normalization of the aldosterone concentration in urine and of urine samples used for western blot analysis, creatinine was measured with an enzymatic assay (Mouse Creatinine Assay Kit, cat.-no. 80350, Crystal-Chem). The reaction was quantified by photometric assessment at 550 nm using an xMark platereader (Bio-Rad).

**Statistics**. Data are shown as mean ± SEM, mean ± SD, or geometric mean ± geometric SD, where $n$ represents the number of animals or cells. Statistical significance was assessed between the groups using the Kruskal–Wallis test with Dunn's multiple comparison test, Mann–Whitney $U$ test, or one-way-ANOVA with Bonferroni multiple comparison test, Prism, GraphPad Software, USA). A $p$ value < 0.05 was considered statistically significant.

Statistical analyses for intracellular $Ca^{2+}$ measurements in Supplementary Fig. 4 were done using "lme4" package (Bates, Maechler & Bolker, 2012) for R to perform a linear mixed effects analysis of the relationship between $Ca^{2+}$ fluorescence and genotype. As fixed effect, we entered genotype into the model; as random effects, we had intercepts for slice and mouse. The "multcomp" package[65] was used for pairwise comparisons (post-hoc test). This model was used to take into account the possible effects of the different slices and mice measured.

**Reporting Summary**. Further information on research design is available in the Nature Research Reporting Summary linked to this article.

## Data availability

The source data underlying Fig. 2a, Supplementary Fig. 2b–d, Supplementary Fig. 10, Table 1 and Supplementary Table 1 are provided as Source Data file. All other data that support the findings of this study are available from the corresponding author upon reasonable request.

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

## Acknowledgements

We thank Patrick Seidler for technical assistance, Ilona Kamer for telemetric blood pressure measurement, Stefanie Weinert for help with the generation of *Clcn2*op mice using homologous recombination in ES cells, and the MDC transgenic facility for blastocyst injection and implantation. This work was supported, in part, by a grant of the Deutsche Forschungsgemeinschaft DFG (Je164/15-1) and the Prix Louis-Jeantet de Médecine to T.J.J., and by the Fondation pour la Recherche Médicale (DEQ20140329556) and by institutional grants from INSERM to M.C.Z.

## Author contributions

T.J.J., M.C.Z., C.G. and I.J.O. designed and evaluated experiments. C.G. generated *Clcn2*op mice, performed western blots, NKCC1 IHC, metabolic cage experiments, and blood electrolyte analysis, and analyzed BP and hormone measurements. C.G. and A.H.S. performed and analyzed adrenal gland histology. I.J.O. performed the electrophysiology experiments and generated and expressed channel mutants; M.B.H.B. performed ClC-2 IHC and qRT-PCR experiments; A.H.S. performed Ca$^{2+}$ measurements; F.L.F.R. and S.B. performed adrenal gland IHC for Cyp11b2 and Dab2; F.L.F.R., S.B. and M.C.Z. analyzed adrenal gland IHC and histology; C.A.H. provided the newly generated NKCC1 antibody and the NKCC1 KO mouse tissue; T.J.J. wrote the manuscript with input from the other authors. The final manuscript was seen and edited by all authors.

## Additional information

**Competing interests:** The authors declare no competing interests.

