## [Peer Review File · Nature Communications]

Reviewers' comments:

Reviewer #1 (Remarks to the Author):

Jentsch and colleagues present a mouse model of primary aldosteronism with a homozygous mutation in the CIC-2 chloride channel. This is a detailed study which extends and clarifies findings of the electrophysiological effects of several mutations associated with familial hyperaldosteronism type II (FH II). They demonstrate that a deletion mutation CLCN2op displays similar electrophysiological properties in *Xenopus* Oocytes as mutations causing FH II (p.Met22Lys, p.Gly24Asp, p.Tyr26Asn). The authors then select the open CIC-2 mutation to produce a homozygous Clcn2op/op model of primary aldosteronism. Thus, the mouse model is not based on a variant associated with the human disorder. In the discussion the authors state "Progress in understanding the pathological mechanisms in detail has been hampered by the absence of mouse models that faithfully replicate the human disease". Then why did the authors select the op/op approach instead of using a CLCN2 mutant which has been described in the disease? The CIC-2op/op mice reproduce some of the biochemical and hormonal abnormalities associated with the human disease although adrenal morphology of the animals appears normal.

"Clcn2op mice provide an excellent mouse model for human PA that will prove useful to explore its pathophysiology ..." This phrase is an exaggeration given the normal adrenal morphology

Table 1: Potassium levels: all pairwise significant differences should be given in addition to differences to +/+. Is the difference significant between female +/op and +/+ mice? If so, could you comment on the difference in potassium levels between +/op compared with +/+ mice in females but not in males?

Fig 2A, please change label to CIC-2 and include beta-actin protein loading control as in supp. Fig 2D.

Fig 2B, densitometric quantification: it isn't clear what was used for the quantification- probably bands from Western blot. This should be stated.

Supp. Fig. 2: Analysis of +/op for all tissues should be included and the reasons for tissue selection given in legend

Supp. Fig 4C and D and legend: the area of the ZG was measured or the thickness of the ZG?

Supp.Fig 5. Statistical analysis: please state there were no differences or provide data.

Supp.Fig 6: Please include +/- data for all measurements

Supp Fig 7: include real-time PCR quantification of CLCN2 of +/- and SF-1-Cre+/- mice

Supp Fig 9: Albuminuria: these were 24 hour urine samples or spot urine samples? Instead of Western blot, a measure of the albumin:creatinine ratio should be given. Was there an increased effect in mice at 52 weeks? Data for aged mice should be provided.

Reviewer #2 (Remarks to the Author):

This manuscript reports the characterization of a knock in mouse model expressing a constitutively open CIC-2 channel. The mouse line generated displays typical features of PA , high serum aldosterone, low renin, hypokalemia and hypertension. The channel alteration does not replicate a genetic variant previously identified in PA, but rather deletes 8 amino acids from the CIC-2 amino-terminus previously identified as a gating domain that harbors two of the CIC-2 PA mutations. However, the CIC-2 open deletion channel produces similar degrees of current enhancement and loss of voltage-dependent gating observed in recordings of disease-causing N-terminus mutations in oocytes and HEK-293 cells. Thus the model provides a demonstration of proof-of-principle that enhanced CIC-2 currents have the potential to produce a PA.

This is a comprehensive set of studies using state of the art techniques to characterize the opCIC-2 mouse.

I have only a few comments.

Major: Analysis

1. Analysis: Of late there has been much discussion in the literature about rigor and reproducibility and the correct assignment of N values in cell and tissue slice experiments. Slices from the same animal are not independent variables as they come from the same animal, and cells

within a slice or cells from one transfection also are not independent variables. It would appear that in the analysis of the calcium imaging data and perhaps in the analysis of some of the electrophysiological data (Fig 1 Supplementary Fig 1, Figure 3) observational variation has been confused with biological variation, as the assigned N in the imaging analysis is 78 cells or greater per genotype rather than N=3 animals per genotype. Proper analysis may change the reported outcomes.

Minor:

2. A gene dose effect may be evident statistically with a linear regression fit of Figure 6a data
3. The strategy to incorporate loxp sites in the mutant channel to allow removal from the adrenal provides a novel rescue of the hyperaldosteronism phenotype. To close the loop, did the removal of the op channel from the adrenal also normalize systolic blood pressure?
4. Deletion of an eight amino acid segment from the amino-terminus is not a human genetic variant. It may produce equivalent currents to channels with the identified single mutant alleles, but it remains unknown if such a deletion disrupts the interactions of ClC-2 with critical binding partners that also may contribute to the measured phenotype. This caveat should be discussed within the context of the statement that the model is “an excellent model for human PA”
5. The repeated use of the word “drastic” as a modifier of depolarization, currents, and aldosterone production is a colloquial not a scientific metric. More appropriate/quantitative adjectives should be used.

Detailed response to reviewers of manuscript NCOMMS-19-04078 "Pathogenesis of hypertension in a mouse model for human CLCN2 related hyperaldosteronism" by Göppner et al.

Reviewer #1 (Remarks to the Author):

Jentsch and colleagues present a mouse model of primary aldosteronism with a homozygous mutation in the CIC-2 chloride channel.

*We have not only studied homozygous $Clcn2^{op/op}$, but also heterozygous $Clcn2^{+/op}$ mice. This is a very important aspect of our study. All crucial data such as membrane potential (and now, newly added, chloride currents) of glomerulosa cells in situ (Fig. 3), cytoplasmic Ca^{2+} -transients of glomerulosa cells from adrenal tissue slices (Fig. 4), expression levels of enzymes involved in aldosterone synthesis (Fig. 5), serum aldosterone and corticosteroid levels and renin activity (Fig. 6), blood pressure (Fig. 7, Suppl. Fig. 9), adrenal morphology (Suppl. Fig. 5), *Dab2* immunostaining (Suppl. Fig. 6) have been determined in both heterozygous and homozygous mice.*

*Whereas heterozygous $+/op$ mice genetically correspond better to the human disease (where the dominant allele is found in a heterozygous state), the homozygous op/op mice are phenotypically closer to the human disease concerning the extent of blood pressure increase (mouse models for hypertension generally have lower increases in blood pressure than humans; the 20 mmHg increase observed in op/op mice is an impressive increase in mouse BP). Further, the more pronounced phenotype of homozygotes facilitates investigation of the underlying pathological mechanisms. Importantly, comparison of $+/op$ and op/op mice yielded interesting insights. We initially suspected that $+/op$ and op/op mice may display the same degree of pathology, since the strong increase in CIC-2 currents by a heterozygously expressed *op* mutant of CIC-2 may suffice to maximally depolarize the ZG cell (close to the Cl equilibrium potential). While the latter point turned out to be true, the disease phenotype (aldo, BP) of heterozygotes was intermediate between op/op and $+/+$ mice because only about half of the ZG cells were depolarized. This is described and discussed in detail in the manuscript.*

This is a detailed study which extends and clarifies findings of the electrophysiological effects of several mutations associated with familial hyperaldosteronism type II in *Xenopus* Oocytes as mutations causing FH II (p.Met22Lys, p.Gly24Asp, p.Tyr26Asn). The authors then select the open CIC-2 mutation to produce a homozygous $Clcn2^{op/op}$ model of primary aldosteronism. Thus, the mouse model is not based on a variant associated with the human disorder. In the discussion the authors state "Progress in understanding the pathological mechanisms in detail has been hampered by the absence of mouse models that faithfully replicate the human disease". Then why did the authors select the op/op approach instead of using a *CLCN2* mutant which has been described in the disease?

We indeed chose to not insert one of the human mutations, but an independent, previously characterized 'artificial' mutation that opens CIC-2. This is not a disadvantage, but rather an advantage as it provides an important proof of principle.

In the first step, we examined all published aldosteronism-associated CLCN2 mutations in two independent expression systems, two-electrode voltage clamp of injected Xenopus oocytes and perforated-patch clamp measurements of transfected HEK cells. This allowed us to compare the effect of all mutations using suitable electrophysiological techniques (the whole-cell measurements presented in the Scholl et al. Nature Genetics paper (2018) are unsuited for CIC-2 because the cytoplasm is dialyzed; hence they did not reveal the 'real' physiological effect of the mutations).

These in vitro studies showed that all described CLCN2 mutations (with the exception of a C-terminal mutation, which most likely represents a silent polymorphism) markedly increase CIC-2 currents ('opened' CIC-2). The three mutations in the N-terminus (p.Met22Lys, p.Gly24Asp, p.Tyr26Asn) and the mutation in the J-K linker (p.Lys362del) increased CIC-2 currents very strongly, whereas the p.Arg172Gln mutation found by Scholl et al. had a milder 'intermediate' phenotype. We thus hypothesized that the more frequent observation of the p.Arg172Gln mutation, its incomplete penetrance, and its presence in multiple generations of the same family can be explained by the less pronounced increase in currents.

These in vitro results suggested to us that the common denominator in these mutations, a strong increase in CIC-2 currents, rather than potential other idiosyncratic effects of the patients' mutations, causes hyperaldosteronism. To test the prediction that any mutation opening CIC-2 would cause PA, we had to choose an independent mutation that opens CIC-2 to the same degree as p.Met22Lys, p.Gly24Asp, p.Tyr26Asn and p.Lys362del. We therefore selected an N-terminal deletion we had thoroughly characterized previously (Gründer..Jentsch, Nature 1992) and which we showed in the present work to have the same electrophysiological effects as those human mutations. Confirming our prediction, this 'op' mutation caused PA in both heterozygous and homozygous mice. Choosing one of the patient's point mutations would most likely have confirmed that it causes PA also in mice, but would not have allowed us to more generally conclude that opening CIC-2 suffices to cause PA. By inserting an artificial mutation that mimics the effect of the majority of PA-associated human CLCN2 mutations, our mice are an excellent model for mutations strongly opening CIC-2 in general. They strongly support the notion that it is simply the opening of CIC-2 which causes disease rather than some functional peculiarity resulting from specific missense mutations at positions 22, 24, and 26 of CIC-2. Moreover, our results show that a mutation in any other chloride channel besides CIC-2 which increases the chloride permeability of ZG cells would cause PA.

We agree with the reviewer that the reason for selecting the op deletion mutant was not sufficiently explained in our manuscript. We therefore changed the text in several places (changes underlined):

Last paragraph of Introduction, lines 90-94:

'Here we found that almost all known PA-associated CLCN2 mutations markedly increased CIC-2 Cl⁻ currents in heterologous expression. The only exception likely represents a silent polymorphism. To test the hypothesis that an increase in CIC-2 currents suffices to cause PA, we generated a novel knock-in mouse model carrying a Clcn2 mutation (Clcn2^{op}) that is not found in human PA, but increases CIC-2 currents to a similar degree as the majority of PA-causing human mutations.'

Results, page 6, lines 154-158:

'With the exception of one functionally silent sequence variant that probably does not cause disease, all PA-associated CLCN2 mutations were associated with strongly increased CIC-2 currents. We

therefore predicted that any other mutation strongly enhancing Cl⁻ currents is able to cause PA. To test this hypothesis, we generated knock-in mice expressing an ‘open’ CIC-2 mutant (op) that has not been found in human disease, but opens CIC-2 to the same degree as human mutations (Suppl. Fig. 2a-c).’

Discussion, end of first paragraph, lines 320-324:

‘The effect of the artificial ‘opening’ mutation in our *Clcn2^{op}* mice strongly suggests that also the human PA-causing *CLCN2* mutations, all of which open CIC-2, cause disease by their increased chloride currents rather than by other mechanisms. It further suggests that mutations increasing other Cl⁻ currents in ZG cells may also provoke aldosteronism.’

*Furthermore, we slightly changed the abstract to stress the comparison of all human *CLCN2* mutations and the choice of an ‘opening’ mutation for our mouse model (lines 24-26):*

‘We now showed that almost all known PA-associated *CLCN2* mutations markedly increase CIC-2 chloride currents and generated knock-in mice expressing a constitutively open CIC-2 chloride channel as a mouse model for PA.’

The CIC-2op/op mice reproduce some of the biochemical and hormonal abnormalities associated with the human disease although adrenal morphology of the animals appears normal.

*The *Clcn2^{op/op}* mice, and to a milder degree the *Clcn2^{+/op}* mice, reproduce the entire biochemical and hormonal abnormalities of the human disease, i.e. high serum (and urine) aldosterone levels, low serum renin, and high blood pressure.*

*Human PA is not always associated with changes in adrenal morphology. There are few data about morphological changes associated with germline mutations and PA. Patients with germline *KCNJ5* mutations exhibit variable adrenal morphology changes, ranging from normal adrenals (Scholl et al, PNAS 2012, PMID 22308486; Mulatero et al, Hypertension 2012, PMID 22203740; Adachi et al, Horm Res Paediatr 2014, PMID 24819081) to adrenals with huge hyperplasia (Geller et al, J Clin Endocrinol Metab 2008, PMID 18505761; Charmandari et al, J Clin Endocrinol Metab 2012, PMID 22628607). No adrenal alterations were described in patients with germline *CACNA1D* mutations (Scholl et al, Nat Genet 2013, PMID 23913001). Patients with germline *CACNA1H* mutations generally do not have adrenal alterations on CT or MRI, except for one reported patient with adrenal cortex hyperplasia (Scholl et al, Elife 2018, PMID 25907736). A recent paper, describing somatic mutations in adrenals resected from patients with idiopathic hyperaldosteronism, shows that in a certain number of cases adrenals from patients with bilateral hypersecretion of aldosterone by adrenal vein sampling do not show adrenal nodules (Omata et al, Hypertension 2018, PMID 30354720).*

*Most patients with *CLCN2* mutations appear to lack morphological changes in the adrenal gland (but, of course, there is no comprehensive histological characterization). The patient we had described (Fernandes-Rosa et al., Nature Genetics 2018) displayed no adrenal gland abnormalities (as examined by CT scan), and among the *CLCN2* patients described by Scholl et al. (Nature Genetics 2018) most lacked detectable morphological changes in the gland. Two of her patients exhibited mild bulkiness of adrenal glands and one had a small adrenal nodule, but with bilateral production of aldosterone. All those patients were from the same family and carried the (‘less strong’) p.Arg172Gln mutation, but other family members with that mutation (including patients) failed to display morphological*

changes. Hence human CLCN2 mutations appear to result only rarely in changes of adrenal morphology, and it is certainly too early to conclude that this is a particular feature of the 'mild' p.Arg172Gln mutation.

In summary, *Clcn2^{op}* mice recapitulate all essential features of human PA. The absence of morphological changes does not argue against the validity of this mouse model.

“Clcn2op mice provide an excellent mouse model for human PA that will prove useful to explore its pathophysiology ...” This phrase is an exaggeration given the normal adrenal morphology

Please see above for the importance of morphological changes – these changes are not consistent features of CLCN2-related PA, nor of other familial forms of PA.

Indeed, we think that these mice ARE an excellent mouse model for human PA. Since the general validity of our *Clcn2^{op}* mice for CLCN2 mutations and the advantages of analyzing hetero- and homozygous mice have already been discussed above, we like to stress here that it is so far the only mouse model for human mutations in genes encoding ion channels or transporters. There are no mouse models recapitulating the human mutations in Ca²⁺- or Na,K-ATPase subunits, nor for the Ca²⁺-channel genes CACN1AD and CACNA1H or the K⁺ channel gene KCNJ5. For the latter channel, only a KO mouse has been analyzed and a female-specific decrease in aldosterone levels was found (Hardege et al., Clinical Science 2018 (PMID:29222092)). However, compared to humans, KCNJ5 shows very low expression in the adrenal gland, and importantly PA-causing KCNJ5 mutations do not ablate channel activity, but endows the mutant channels with a depolarizing Na⁺ conductance.

Several KO mouse models for TASK K⁺ channels have been reported (Heitzmann et al, EMBO J 2008, PMID 18034154; Penton et al, Endocrinology 2012, PMID 22878402; Davies et al, Proc Natl Acad Sci 2008, PMID 18250325; Guagliardo et al, Hypertension 2012, PMID: 22493079). Mice variably display hypertension and adrenal morphological abnormalities, but these channel genes are not mutated in human PA.

Thus, we believe that we do not exaggerate with our statement (*Clcn2^{op}* mice are an excellent mouse model').

To clarify this point further, we have added a sentence at the end of Introduction, lines 100-102:

'Although human PA can be caused by mutations in 6 different genes coding for ion transport proteins, the present mice are the first animal model for mutations in such a gene.'

Table 1: Potassium levels: all pairwise significant differences should be given in addition to differences to +/+. Is the difference significant between female +/-op and +/+ mice? If so, could you comment on the difference in potassium levels between +/-op compared with +/- mice in females but not in males?

The potassium levels of females +/-op compared with +/+ is NOT significant (adjusted p-value = 0.3098). We only indicate significance when it results in p<0.05, as mentioned in Materials and Methods.

However, we would like to mention here that sexual dimorphism is quite common with adrenal phenotypes, see e.g. (Hardege et al., *Clinical Science* 2018 (PMID:29222092); cited above for *Kcnj5* KO) and Heitzmann et al., *EMBO J* 2008 (PMID:18034154) for the *TASK1* KO, and Dumontet et al., *JCI Insight* 2018 (PMID:29367455) for PKA-driven differentiation. For instance, we observed that expression of *CIC-2* in the adrenal gland is somewhat higher in female mice, see Fig. 2a.

Fig 2A, please change label to *CIC-2* and include beta-actin protein loading control as in Suppl. Fig 2D.

*The Figure is labelled correctly – *Clcn2* refers to the genotypes at the left (indicated by *-/-*, *+/+* etc), the *CIC-2* protein band is labeled correctly with *CIC-2*.*

As suggested, we have now added the actin controls to Fig. 2a. Actin controls for membrane proteins preparations, however, are slightly problematic as actin is not an intrinsic membrane protein. We confide more in biochemical protein quantification which we have done for each Western blot, loading equivalent amounts of proteins.

Fig 2B, densitometric quantification: it isn't clear what was used for the quantification- probably bands from Western blot. This should be stated.

*Yes, these were the protein bands. 'protein bands' has been added to the legend. It reads now (line 698): 'Densitometric quantification of protein bands to determine *CIC-2* expression...'*

Suppl. Fig. 2: Analysis of *+/op* for all tissues should be included and the reasons for tissue selection given in legend

*We prefer not to show also *+/op* tissue in Suppl. Fig. 2d – in view of the fact that we have no difference in expression between *op/op* and *+/+*, no difference to *+/op* is expected either. Moreover, Western analysis of *+/-* tissue is shown in main Fig. 2a for the adrenal gland, the tissue we are concerned with here.*

*Indeed, we have done Western blots of *+/-* tissue also for the other tissues, which we show here as Fig. R1 for the reviewer only. There is no difference in *CIC-2* protein expression in *+/op* mice, as expected. We don't deem it necessary to include the Figure in Supplement, but could do so if the reviewer insists on doing this.*

Fig. R1: Representative Western blot for *CIC-2* of membrane fractions isolated from whole brain, cerebellum and distal colon of *Clcn2*^{+/+} (*+/+*), *Clcn2*^{+/*op*} (*+/*op**), *Clcn2*^{*op/op*} (*op/op*) mice. Antibody specificity was verified with *Clcn2*^{-/-} (*-/-*) tissue. Equal amounts of protein were loaded with β -actin serving as loading control.

We have chosen colon and brain for comparison because CIC-2 is expressed to comparatively high levels in those tissues (the expression in colon is mentioned in the manuscript in the context of the immunohistochemistry we used to show that the *op* mutation does not change the localization of the protein, and in our experiment aimed at determining a mono-allelic expression of CIC-2). We have now added the rationale for using these tissues in the legend to Suppl. Fig. 2:

‘Brain and colon were chosen because they display robust expression of CIC-2 and have been studied previously (and in this work) by techniques that included CIC-2 immunohistochemistry¹⁻³.’

- 1 Blanz, J. *et al.* Leukoencephalopathy upon disruption of the chloride channel CIC-2. *J Neurosci* **27**, 6581-6589 (2007).
- 2 Hoegg-Beiler, M. B. *et al.* Disrupting MLC1 and GlialCAM and CIC-2 interactions in leukodystrophy entails glial chloride channel dysfunction. *Nature communications* **5**, 3475, doi:10.1038/ncomms4475 (2014).
- 3 Catalán, M. *et al.* CIC-2 in guinea pig colon: mRNA, immunolabeling, and functional evidence for surface epithelium localization. *Am J Physiol Gastrointest Liver Physiol* **283**, G1004-G1013 (2002).

Suppl. Fig 5C and D and legend: the area of the ZG was measured or the thickness of the ZG?

The ZG area was measured. To specify this, the legend of Suppl. Fig. 5c and d now states: ‘Quantification of the area of the zona glomerulosa (ZG)’. In the Materials and Methods section (lines 508-509) we now included: ‘Zona glomerulosa area was quantified relative to the whole adrenal gland area using ImageJ software’.

Suppl. Fig 6. Statistical analysis: please state there were no differences or provide data.

Suppl. Fig 6 legend now states: ‘No significant differences were found by performing statistical analysis using one-way-ANOVA, followed by Bonferroni multiple comparison test.’

Suppl. Fig 7: Please include +/-op data for all measurements

Owed to limitations in the number of mice we were allowed to study by the authorities, we have not performed this analysis for heterozygous animals. Inclusion of new data would require the application for a new animal experiment permit, which could take 4-6 months until approval (with uncertain success, as we have all the relevant data for +/+ and -/- mice, with no significant new insight expected when repeating it for heterozygous mice). As summarized in detail above in our first answer to the

reviewer, we have obtained all crucial parameters for +/+, +/op, and op/op mice and are convinced that Suppl. Fig. 7 does not need these additional data. These are not needed for the conclusions of our paper.

Supp Fig 8: include real-time PCR quantification of CLCN2 of +/+ and SF-1-Cre+op/op mice

We believe that the examination of CLC-2 expression by immunohistochemistry in the conditional KO mouse model, as shown in Suppl. Fig. 8, is much preferable to real-time PCR, because the Cre line deletes specifically in adrenal cortex (and not in medulla). IHC shows the specificity of deletion and confirms again the specificity of our CLC-2 antibody. qRT-PCR of the entire adrenal gland, by contrast, would be unable to show the extent of deletion in the cortex – the relevant site of expression for our study.

Nonetheless, following the request of the reviewer, we have now performed also qRT-PCR analysis of CLC-2 expression (Fig. R2 for reviewers only). As expected, we find an incomplete decrease in CLC-2 mRNA because the channel is also found in the medulla. We prefer not to include this experiment into the manuscript, as the investigation by IHC is much better.

Fig. R2: Quantitative RT-PCR analysis of *Clcn2* mRNA expression in adrenal glands from 13-26 week-old male and female *Clcn2*^{+/+} (+/+, circles) and SF1-Cre+; *Clcn2*^{op/op} (SF1-Cre+ op/op, triangles) mice, *n* ≥ 3 (each point represents one animal). No significant difference was found using Mann-Whitney U test. Error bars, geometric mean ± geometric SD.

Supp Fig 10: Albuminuria: these were 24 hour urine samples or spot urine samples? Instead of Western blot, a measure of the albumin:creatinine ratio should be given. Was there an increased effect in mice at 52 weeks? Data for aged mice should be provided.

As stated in Methods (line 477), the urine samples represent 24h samples from mice kept in metabolic cages for other experiments (Suppl. Fig 7d). As stated in the Figure Legend, the amount of urine loaded was adjusted to creatinine concentration and therefore already represents the normalization requested by the reviewer.

To immediately make clear that the urine has been collected over 24 hours, we now state this not only in Methods, but also in the legend to Suppl. Fig. 10: '24h-urine samples from ...'.

We have provided the proteinuria data only to show the potential of our mouse model for future studies, and believe that they are interesting and promising enough to be included. These data show that our mouse model will be useful to study medium- to long-term consequences of high blood pressure and high aldosterone in our mouse model. A complete investigation of such secondary effects is clearly beyond the scope of our paper which is concerned with the elucidation of the mechanism by which activating CIC-2 mutations lead to primary aldosteronism and hypertension. Investigation of the time course of proteinuria would also require breeding more mice and applying for a new animal experimentation permit, which would take many months.

*To emphasize the future potential of our mouse model, we have added a sentence at the end of the paragraph describing proteinuria: 'Although end organ damage was not explored further, this observation suggests that *Clcn2*^{op/op} mice might prove useful to investigate secondary tissue damage in PA.'*

Reviewer #2 (Remarks to the Author):

This manuscript reports the characterization of a knock in mouse model expressing a constitutively open CIC-2 channel. The mouse line generated displays typical features of PA, high serum aldosterone, low renin, hypokalemia and hypertension. The channel alteration does not replicate a genetic variant previously identified in PA, but rather deletes 8 amino acids from the CIC-2 amino-terminus previously identified as a gating domain that harbors two of the CIC-2 PA mutations. However, the CIC-2 open deletion channel produces similar degrees of current enhancement and loss of voltage-dependent gating observed in recordings of disease-causing N-terminus mutations in oocytes and HEK-293 cells. Thus the model provides a demonstration of proof-of-principle that enhanced CIC-2 currents have the potential to produce a PA.

This is a comprehensive set of studies using state of the art techniques to characterize the opCIC-2 mouse.

We thank the reviewer for appreciating the quality of our data and the conclusions of our work.

*The reviewer correctly points out that we have not inserted a variant previously identified in human PA, but a deletion, and states that this provides a proof-of-principle. We would like to refer this reviewer to our discussion of the advantage of using an 'artificial' mutation which we wrote in response to the critical question of reviewer 1. Indeed, using this artificial deletion that is not found in human disease, we tested (successfully) the prediction that *CLCN2* mutations cause PA by strongly increasing CIC-2 currents. Our work shows in a general way that creating a large Cl conductance in ZG cells, as done in our mouse model and with disease-causing *CLCN2* mutations, causes PA. As stated in the response to reviewer 1, we now have emphasized our approach and its rationale at various places in the text.*

I have only a few comments.

Major: Analysis

1. Analysis: Of late there has been much discussion in the literature about rigor and reproducibility and the correct assignment of N values in cell and tissue slice experiments. Slices from the same animal are not independent variables as they come from the same animal, and cells within a slice

or cells from one transfection also are not independent variables. It would appear that in the analysis of the calcium imaging data and perhaps in the analysis of some of the electrophysiological data (Fig 1 Supplementary Fig 1, Figure 3) observational variation has been confused with biological variation, as the assigned N in the imaging analysis is 78 cells or greater per genotype rather than N=3 animals per genotype. Proper analysis may change the reported outcomes.

This comment is well taken, and we have corrected the paper accordingly. This required some new experiments.

We have always indicated not only the number of cells, but also the number of animals. For perforated patch-clamp measurements of ZG cells in situ, we had – because of the difficulty of the technique – mostly only one cell per animal (see below), but the discrepancy is much larger for the Ca-measurements on slices, where we had only 3 animals per genotype.

Following the advice of the reviewer, we have now replicated the Ca²⁺ measurements in additional 3 independent experiments per genotype (3 more mice per genotype) and have calculated the significance of the differences using means of each measurement (Fig. 4c). We have added a new Suppl. Fig 4 in which we show the individual measurements for each cell from these (now) six mice per genotype, color-coded for each mouse. We believe it is important to not just show a mean for each mouse, but also the scatter of values within each mouse. For the electrophysiological experiments with slices, we have added more data points (Fig. 3d) and have newly included Cl⁻ current measurements of ZG cells of all three genotypes (Fig. 3c). We have also added more data points (each point corresponding to one mouse) for Suppl. Fig. 8 that describes aldosterone synthase mRNA levels, aldosterone concentration, and renin activity for our conditional SF1-Cre; Clcn2^{op/op} mouse model.

Owed to the difficulty in obtaining stable perforated-patch recordings of ZG cells in situ, most data points in Fig. 3 a-c and d represent a measurement from a different mouse. Only in ~20% of the cases we were able to measure 2 cells from the same mouse. The following Table gives the exact numbers:

Fig 3a-c (chloride currents)	# cells measured	# mice used
+/+	7	7
+/op	9	8
op/op	6	6
Fig 3d (membrane potential)		
+/+	9	9
+/op	18	12
op/op	10	8

The numbers of cells and mice are now specified in the legend to Fig. 3 (line 708-709)

When two cells were measured per +/op mouse, there seemed to be a random distribution of depolarized or hyperpolarized V_m, i.e. within a given +/op mouse both hyperpolarized and depolarized ZG cells were found.

Minor:

2. A gene dose effect may be evident statistically with a linear regression fit of Figure 6a data

Thank you for this suggestion. We have now performed this analysis, which is shown below in Fig. R2 for the reviewers only.

Fig. R2: Linear regression fit of data in Fig. 6a

This analysis indeed indicates a statistically significant gene dosage effect, but we prefer not to show this additional test in our manuscript.

3. The strategy to incorporate loxp sites in the mutant channel to allow removal from the adrenal provides a novel rescue of the hyperaldosteronism phenotype. To close the loop, did the removal of the op channel from the adrenal also normalize systolic blood pressure?

We have not analyzed blood pressure in these mice, which would require an extensive series of new experiments for which we would need a new animal experimentation permit from the authorities (this would take in total 6 months or more). Moreover, these experiments are clearly beyond the scope of our work, which focuses on the mechanism by which 'open' CIC-2 channels cause hyperaldosteronism. The deletion of the dominant allele in ZG cells shows clearly that the effect is cell-autonomous for these cells. Importantly, as mentioned above, we now added newly obtained data points from more animals (Suppl. Fig. 8 b-d). These confirm our conclusions.

4. Deletion of an eight amino acid segment from the amino-terminus is not a human genetic variant. It may produce equivalent currents to channels with the identified single mutant alleles, but it remains unknown if such a deletion disrupts the interactions of CIC-2 with critical binding partners that also may contribute to the measured phenotype. This caveat should be discussed within the context of the statement that the model is "an excellent model for human PA"

We agree that the deletion could change interactions with other proteins, but the same argument can also be made for the human point mutants. In our paper we have investigated both the protein abundance as well as the subcellular localization (in colon) of the deletion mutant in mice to partially address this question experimentally, and did not find significant changes. Of course, this cannot exclude interactions that have other effects.

Most importantly, the fact that mice carrying the deletion display PA similar to the patients who have a number of different point mutations suggests that altered protein-protein interaction, either with our deletion or the human point mutations, do not play a significant role in the pathology – it comes down to the increased Cl⁻ conductance.

Nonetheless, we agree with the reviewer that it is good scientific practice to acknowledge the possibility of such an effect, and have inserted on page 7, lines 173-175:

‘This suggests that the deleted amino-terminal residues do not interact with proteins involved in CIC-2 protein trafficking and localization, but as with any other mutant, we cannot strictly exclude a loss or gain of other interactions’.

5. The repeated use of the word “drastic” as a modifier of depolarization, currents, and aldosterone production is a colloquial not a scientific metric. More appropriate/quantitative adjectives should be used.

We had used the qualifier ‘drastic’ only for >10-fold changes. Following the suggestion of the reviewer, we have now replaced it in several places by e.g. ‘roughly 10-fold increase’ to give a more quantitative notion in the text itself, and by ‘marked’ or ‘strong’ at other places since we think that repeating the more quantitative descriptors many times would rather disturb the flow of the text. These changes are highlighted in the marked copy we have uploaded.

We thank both reviewers for thoroughly reading our manuscript and for their useful suggestions.

Additional changes (not requested by the reviewers) made to improve the manuscript:

(1) To further improve the reliability of our data, we have increased the number of measurements for several experiments:

(a) We have increased the number of measurements of the resting voltage V_m of ZG cells in heterozygous $Clcn2^{+/op}$ mice in Fig. 3d. This, of course, corresponds to analysis of new mice.

(b) We have increased the number of data points for Cyp11b2, aldosterone and renin from SF1-Cre; $Clcn2^{op/op}$ mice (Suppl. Fig. 8b-d).

The increase in the number of measurements further supports our conclusions.

(2) Moreover, we have added a new important experiment that is relevant for the interpretation of the bimodal distribution of V_m in heterozygous mice: We now also measured the Cl⁻ currents of these cells in the perforated patch-clamp technique. These currents also show a bimodal distribution (new panel c of Fig. 3), virtually eliminating the previously discussed possibility that the bimodal distribution of V_m is caused by non-linear properties of ZG cells that can push (or not) V_m to one of

two values in face of a depolarizing current of same magnitude in all ZG cells. It rather appears that the expression of ClC-2 currents is already bimodal, being present in only about half of the ZG cells in heterozygous mice even though the channel is expressed (as determined by IHC) in all cells.

These new results are now described in results and the implications for the bimodal distribution of V_m are discussed on page 14 line 392-394:

'However, the fact that also ClC-2 currents appeared to have a bimodal distribution in ZG cells (Fig. 3c) rather points to an upstream factor controlling the amplitude of these currents'.

(3) *We have performed new experiments in which we ascertained the specificity of the IHC for NKCC1 in the adrenal gland by using $Nkcc1^{-/-}$ adrenal glands. This important control has now been added to Fig. 2d.*

REVIEWERS' COMMENTS:

Reviewer #1 (Remarks to the Author):

No further comments

Reviewer #2 (Remarks to the Author):

The authors have thoughtfully considered all comments in the initial review and have added new data, new analyses and new text to the manuscript addressing all concerns.

Minor

I would only suggest that more representative IHC images be included in Supplemental Figure 6. The op/po adrenal slice images do not seem to reflect the quantification provided, but rather they suggest that there is some zG cell de-differentiation in the op/op mouse line.

NCOMMS-19-04078A-Z 'Pathogenesis of hypertension in a mouse model for human *CLCN2* related hyperaldosteronism' by Göppner *et al.*

Response to reviewers

We thank both reviewers for taking their time to re-evaluate our work and are glad that they are satisfied with the revised version.

As minor point, reviewer 2 pointed out that we should provide a more representative image for the Dab2 IHC in Supplementary Figure 6. We have changed the figure accordingly.